# Linguistic Nepotism: Trading-off Quality for Language Preference in Multilingual RAG

**Dayeon Ki** [* 1]  **Marine Carpuat** [1]  **Paul McNamee** [2]  **Daniel Khashabi** [2]
**Eugene Yang** [2]  **Dawn Lawrie** [2]  **Kevin Duh** [2]

## Abstract

Multilingual Retrieval-Augmented Generation (mRAG) systems enable language models to answer knowledge-intensive queries with citation-supported responses across languages. Despite their growing use, an open questions is whether the mixture of different document languages impacts generation and citation behavior in *unintended* ways. To investigate this, we introduce a controlled methodology using model internals to measure language preference while holding other factors such as document relevance constant. Across eight languages and six open-weight models, we find that models preferentially cite English sources when queries are in English, with this bias amplified for lower-resource languages and for documents positioned mid-context. More crucially, we find that models sometimes trade-off document relevance for language preference, indicating that citation choices are not always driven by informativeness alone. Our findings shed light on how language models leverage multilingual context and influence citation behavior.[1]

## 1. Introduction

Retrieval-Augmented Generation (RAG) systems have become a core component of modern Large Language Model (LLM) pipelines, enabling models to answer knowledge-intensive queries by supplementing their limited parametric knowledge with external information (Lewis et al., 2020; Karpukhin et al., 2020; Gao et al., 2024). Given that over 50% of digital content is produced in languages other than

English (Statista, 2025), recent work has extended these systems to multilingual RAG (mRAG) settings, which handle queries and documents in languages beyond English (Chirkova et al., 2024; Wu et al., 2024).

Despite recent advances, prior work highlights a key challenge in mRAG systems: **language preference**—a systematic tendency of models to favor sources written in certain languages during generation (Park & Lee, 2025). Understanding this behavior is crucial, as citation patterns shape both the information users see and the languages prioritized in multilingual knowledge access.

Existing approaches to measuring language preference, however, often fail to capture citation correctness. In *short*-form mRAG, preference has been estimated through information overlap (Sharma et al., 2025) or embedding similarity (Park & Lee, 2025), which do not directly account for correctness. In *long*-form mRAG, where outputs contain in-line citations (Zheng et al., 2025; Xu & Peng, 2025), preference has been measured by comparing citation frequencies against the language distribution of retrieved documents. Yet, this signal is coarse and confounded by the relevance and informativeness of multilingual sources ($C_1$). Moreover, in-line citations are prone to hallucinations (Gao et al., 2023; Zhang et al., 2025b), making it unclear whether observed preferences reflect true attribution or spurious citations ($C_2$).

To address both of these challenges, we propose a controlled methodology for measuring language preference using model internal metrics (illustrated in Figure 1). Here, we first construct a synthetic multi-parallel dataset of relevant documents, which allows us to isolate the effect of language while controlling for other factors such as document content and relevance (Step 1+2; addresses $C_1$). Citation correctness is then verified through a two-step filtering process (Step 3; addresses $C_2$) (§3.1). Next, we compare the accuracy of next token citation predictions (e.g., predicting "2" for document ID 2) while varying the language of the same cited document and keeping other variables fixed, including the language of remaining documents, document positions in the input context, and the query language (Step 4). Differences in citation accuracy between languages indicate a preference for the higher-accuracy language (§3.2).

---

*Work done while visiting at Johns Hopkins University.

[1]University of Maryland [2]Johns Hopkins University. Correspondence to: Dayeon Ki <dayeonki@umd.edu>.

*Proceedings of the 43rd International Conference on Machine Learning*, Seoul, South Korea. PMLR 306, 2026. Copyright 2026 by the author(s).

[1]Code and data are released at https://github.com/dayeonki/lang_preference.

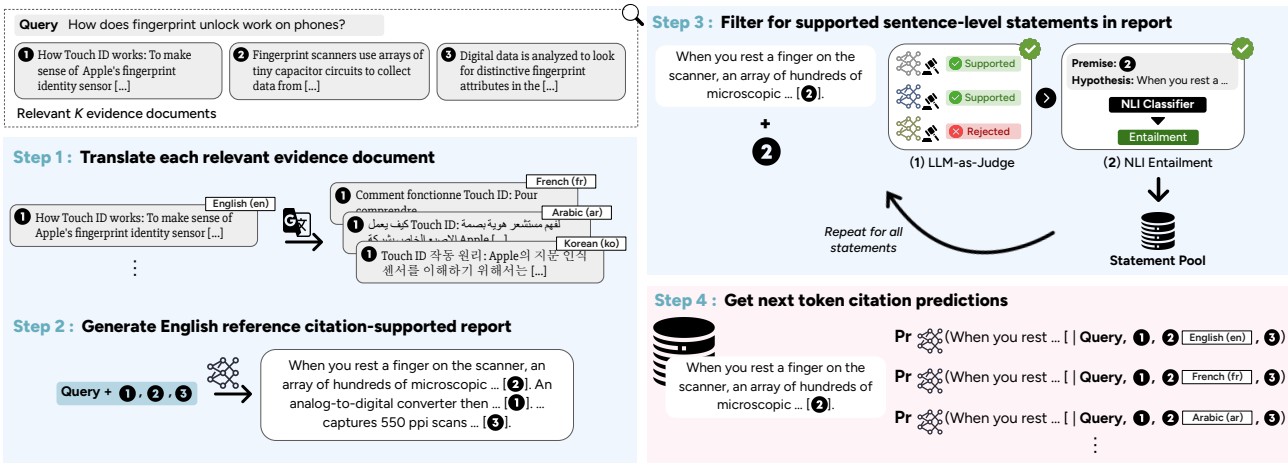

Figure 1. **Overview of our approach for measuring language preference.** We show both synthetic data generation and measurement method. Given an English query $q$ and its $K$ relevant evidence documents $D_{en}$, we first translate the documents into multiple languages $D_{fr}, D_{ar}, D_{ko} \ldots$ (Step 1). We then generate a *reference* citation-supported report $r$ for each query using $q$ and $D_{en}$ (Step 2). The report $r$ consists of sentence-level statements $s_i$, each paired with a single citation ID $c_i$. For each $r$, we retain only statements that are verified (Step 3). Language preference is detected when the next token prediction accuracy for the correct citation ID decreases as the language of the cited document is varied (Step 4).

Using this setup across eight languages and six open-weight models, we address the overarching question: Do models preferentially cite documents in certain languages during long-form mRAG? To further inform building more robust mRAG systems, we empirically address three key questions: (*i*) What factors amplify language preference? (*ii*) What role does the query language play here? and (*iii*) Is citation behavior driven more by document relevance or language?

Our main findings can be summarized as follows:

- **Evidence of strong English preference:** Across all tested models, we find a pronounced tendency to cite English documents when the query is in English. This preference amplifies when: (1) the cited document is in a lower-resource language (e.g., Bengali, Swahili), or (2) the cited document appears in the middle of the input context (§5).
- **Language preference towards query language:** We show that language preference extends beyond English: models favor citing evidence documents written in the query language (§6).
- **Language outweigh relevance:** Last but not least, we show that models frequently cite English documents even when they are *irrelevant* to the user query, suggesting that language itself exerts a stronger influence than document relevance in long-form mRAG (§7).

## 2. Related Work

**Multilingual RAG.** A growing body of work has examined that Large Language Models (LLMs) are prone to hallucinations, especially in knowledge-intensive tasks

(Augenstein et al., 2024; Huang et al., 2025a). Retrieval-Augmented Generation (RAG) mitigates this by retrieving external knowledge sources and incorporating them into generation (Chen et al., 2024b; Gao et al., 2024). While early RAG systems largely focused on processing English queries and sources, recent research has extended these methods to multilingual RAG (mRAG), enabling retrieval and generation across a wider range of languages (Asai et al., 2022). Prior mRAG studies primarily examine the effects of query language (Chirkova et al., 2024), the language of relevant or irrelevant evidence documents (Wu et al., 2024; Qi et al., 2025; Liu et al., 2025), document ordering (Ranaldi et al., 2026), and prompting strategies (Ranaldi et al., 2025) on performance. However, due to cost efficiency and scalability (Saad-Falcon et al., 2024; Es et al., 2024), most of this work targets *short*-form mRAG, where the output is a brief answer to a factoid-style query (e.g., "What is the capital of France?"). In contrast, we focus on *long*-form mRAG, where models are asked to generate citation-supported reports in response to open-ended queries (e.g., "How does fingerprint unlock work on phones?").

**Long-form (m)RAG.** Long-form RAG systems build upon prior work on long-form question answering datasets (Stelmakh et al., 2022), which generate paragraph level, citation-supported responses for complex, knowledge-intensive queries (Zhao et al., 2024; Wei et al., 2024; Ju et al., 2025; Zhang et al., 2025a). Despite evaluating models on long-form outputs is notoriously challenging (Qi et al., 2024), it is also increasingly important as this setup better mirrors how humans naturally interact with search engines (Khashabi et al., 2021), making such systems more easily

integrable into search-based workflows like Deep Research platforms (Huang et al., 2025b; Zheng et al., 2025). Similarly, we use a long-form RAG dataset, Explain Like I'm Five (ELI5) (Fan et al., 2019), to measure language preference in mRAG.

**Language Preference.** Language preference describes a systematic tendency for models to favor sources in certain languages over others. This preference largely arises from differences in training data distribution, tokenization methods, and resource availability (Wu et al., 2024; Sharma et al., 2025; Shen et al., 2024). Such preference manifests at both the retrieval and generation stages. On the retrieval side, prior work shows that Multilingual Information Retrieval (MLIR) systems tend to favor high-resource languages (e.g., English) while under-representing sources in lower-resource languages, which can degrade retrieval quality (Telemala & Suleman, 2022; Yang et al., 2024; Amiraz et al., 2025) and introduce inconsistencies in generation (Chataigner, Cléa and Taïk, Afaf and Farnadi, Golnoosh, 2025). On the generation side, language models have been found to more effectively utilize sources written in specific languages (Park & Lee, 2025). Existing studies on short-form mRAG measure this by querying models in various languages and measuring information overlap (Sharma et al., 2025) or embedding similarity (Park & Lee, 2025) between outputs and reference answers. In long-form setting, prior work approximates language preference by comparing citation rates against the distribution of available documents per language, where over-representation signals bias (Li et al., 2025). We build on this line of measuring language preference in long-form mRAG, but through a more controlled experimental setup using model internal metrics.

# 3. Measuring Language Preference in Long-form mRAG

Our goal is to measure whether LLMs systematically prefer citing evidence in some languages over others. To do this, we need (*i*) a multilingual dataset of queries with parallel evidence documents and verifiable citation-supported reports (§3.1), and (*ii*) a measurement method that compares citation accuracy when the same document is presented in different languages (§3.2) as shown in Figure 1. All relevant prompts are provided in Appendix A.

## 3.1. Synthetic Data Generation

**Step 1: Evidence Document Translation.** Let $\mathcal{D}_{en} = \{d_1, \ldots, d_K\}$ denote the set of $K$ relevant evidence documents in English associated with a query $q$. Since no parallel long-form mRAG datasets are publicly available, we construct multilingual variants $\mathcal{D}_{\ell_{target}}$ for each target language $\ell_{target} \in \mathcal{L}_{target}$ using Machine Translation (MT). If

$MT_\ell$ denote a translation function into language $\ell$, we obtain $\mathcal{D}_\ell = \{MT_\ell(d_1), \ldots, MT_\ell(d_K)\}$. In our experiments, $MT_\ell$ is implemented using Google Translate API. Despite the challenges of translating long-context documents (Wang et al., 2023; 2025b), the translation quality remains reasonable, with average COMET[2] quality estimation scores of 0.541. Per-language scores are reported in Appendix D.1.

**Step 2: Reference Report Generation.** For each query $q$ with associated English evidence document set $\mathcal{D}_{en} = \{d_1, \ldots, d_K\}$, we generate a *reference* citation-supported report using a strong LLM $\mathcal{M}_{gen}$. We select OpenAI o3 as $\mathcal{M}_{gen}$, since its outputs were rated highest by human evaluators in SciArena (Zhao et al., 2026), a benchmark assessing long-form report generation and citation quality. The generated report is: $r = \mathcal{M}_{gen}(q, \mathcal{D}_{en})$.[3] We segment $r$ into $n$ sentence-level statements: $r = (s_1, [c_1], \ldots, s_n, [c_n])$, where $s_i$ is the $i$-th statement, and $c_i \in \{1, \ldots, K\}$ is the citation ID of the evidence document $d_{c_i} \in \mathcal{D}_{en}$ that $\mathcal{M}_{gen}$ cites as supporting $s_i$. By construction, $c_i$ denotes the citation token appearing in the report after $s_i$.

**Step 3: Statement Pool Construction.** Long-form generation with citations is prone to hallucination, with LLMs often introducing factual errors (Ji et al., 2023) or misattributing information to incorrect evidence (Gao et al., 2023; Magesh et al., 2024; Zhang et al., 2025b). To ensure that only verifiably supported statements are retained for evaluation, we apply a two-stage filtering pipeline to the set of statement-citation pairs $\{(s_i, c_i)\}_{i=1}^n$ from **Step 2**. We perform filtering only if $|c_i| = 1$ (i.e., statements with exactly one citation). First, the LLM-as-Relevance-Judge identifies statements whose cited document is deemed most relevant by the majority of judges, capturing correctness in the statement $\rightarrow$ cited document direction.[4] Second, the NLI entailment check verifies that the cited document actually entails the information in the statement, capturing in the cited document $\rightarrow$ statement direction.

**(1) LLM-as-Relevance-Judge:** Let $\mathcal{M}_{judge} = \{m_1, m_2, m_3\}$ be the set of judge models that rank highest on the SciArena benchmark (OpenAI o4 mini, QWEN-3 32B (Yang et al., 2025), and Gemini 2.5 Pro). Each judge $m \in \mathcal{M}_{judge}$ is prompted with statement $s_i$ and the full evidence document set $\mathcal{D}_{en}$ to return the index of the most relevant document $j_m(s_i, \mathcal{D}_{en})$. Here, $j_m$ implements a relative selection task over all $\mathcal{D}_{en}$ (i.e., "Which document best supports the statement?"), rather than an absolute binary support judgment (i.e., "Does this document support the statement?"), following findings that comparative framing improves LLM

---

[2]Unbabel/wmt22-cometkiwi-da
[3]Reports contain an average of 148.5 words over 4.9 sentences.
[4]Prior work shows that LLMs provide precise relevance assessments (Ma et al., 2024; Sun et al., 2023).

| Language | LLAMA-3.1 8B | QWEN-3 8B | AYA23 8B | QWEN-3 14B | GEMMA-3 27B | LLAMA-3.3 70B |
|---|---|---|---|---|---|---|
| **English** | 67.4 | 62.6 | 60.0 | 83.0 | 86.2 | 85.9 |
| **French** | 62.9 (-4.49) | 48.4 (-14.2)*** | 48.5 (-11.5)*** | 76.0 (-7.04)*** | 79.0 (-7.21)** | 77.4 (-8.50)*** |
| **Russian** | 62.1 (-5.30)* | 50.4 (-12.2)*** | 48.1 (-11.9)*** | 74.8 (-8.17)*** | 77.1 (-9.12)*** | 74.5 (-11.4)*** |
| **Spanish** | 62.1 (-5.32)* | 51.9 (-10.7)*** | 49.1 (-10.9)*** | 77.4 (-5.61)* | 80.2 (-6.04)** | 76.0 (-9.90)*** |
| **Korean** | 61.7 (-5.68)* | 49.7 (-12.9)*** | 42.2 (-17.8)*** | 70.3 (-12.7)*** | 77.5 (-8.71)*** | 69.2 (-16.7)*** |
| **Chinese** | 59.9 (-7.51)* | 49.2 (-13.4)*** | 46.3 (-13.7)*** | 73.5 (-9.49)*** | 75.4 (-10.8)*** | 74.1 (-11.8)*** |
| **Arabic** | 59.5 (-7.91)** | 47.6 (-15.0)*** | 43.2 (-16.8)*** | 72.6 (-10.4)*** | 78.4 (-7.82)*** | 67.3 (-18.6)*** |
| **Bengali** | 56.6 (-10.8)*** | 41.3 (-21.3)*** | 27.2 (-32.8)*** | 65.4 (-17.6)*** | 77.9 (-8.33)*** | 68.8 (-17.1)*** |
| **Swahili** | 53.0 (-14.4)*** | 30.4 (-32.2)*** | 22.4 (-37.6)*** | 54.7 (-28.3)*** | 74.0 (-12.2)*** | 67.3 (-18.6)*** |

*Table 1.* **Citation accuracies (%) by model and language.** We present mean accuracy values **Acc**$^{(\ell)}$ with $\Delta(\ell_{\text{target}})$ in subscript. Pairwise two-sided $t$-tests with Bonferroni correction are performed to compare accuracy between English and the target language, with null hypothesis as the mean citation accuracy being equal across languages. *: significant with $p < 0.05$; **: $p < 0.01$; ***: $p < 0.001$; non-marked: not statistically significant. Color coding indicates the magnitude of $\Delta(\ell_{\text{target}})$: largest, second largest, others. Columns: increasing model size; rows: decreasing $\Delta(\ell_{\text{target}})$ (of first model). All models consistently show English preference.

evaluation accuracy (Godfrey et al., 2025). The total number of judges selecting the cited document $d_{c_i}$ is:

$$\mathbf{votes}(s_i, c_i) = \sum_{m \in \mathcal{M}_{\text{judge}}} \mathbb{1}(j_m(s_i, \mathcal{D}_{\text{en}}) = c_i) \quad (1)$$

We retain $s_i$ if when the majority of judges agree on the correct judgment: $\mathbf{votes}(s_i, c_i) \geq 2$.

**(2) NLI Entailment:** We use an off-the-shelf Natural Language Inference (NLI) classifier $\phi(\text{premise}, \text{hypothesis})$[5], which outputs 1 if the premise entails the hypothesis, and 0 otherwise. In our setting, $d_{c_i}$ is the premise and $s_i$ the hypothesis. We retain $s_i$ if $\phi(d_{c_i}, s_i) = 1$. This is in accordance with the Attributable to Identified Sources (AIS) framework (Rashkin et al., 2023). Both our human annotation results in Appendix C.1 and end-to-end evaluator results in Appendix C.2 show high agreement with the automatic NLI filtering judgments.

In practice, the LLM-as-Relevance-Judge and NLI Entailment filtering stages achieve retain rates of 90.35% and 96.12%, respectively. The final pool consists of 792 statements that pass both filters,[6] ensuring that the correctness of each citation used for evaluation is reliably verified.

### 3.2. Measurement Method

**Step 4: Next Token Prediction Analysis.** Intuitively, if the model predicts the correct citation token when the cited document is in English than in other languages, this indicates a preference for English. To quantify this, for each verified statement-citation pair $(s_i, c_i)$, we measure whether the model predicts $c_i$ as the top-1 next token.

We first construct a citation prediction prompt ending in the form: $x_i = s_i$ [, where [ signals the start of the citation. To

---
[5]`mDeBERTa-v3-base-xnli-multilingual-nli`
[6]On average, each verified statement contains 33.7 words.

test for language preference for English, we define the set of evaluation languages as $\mathcal{L}_{\text{eval}} = \{\text{en}\} \cup \mathcal{L}_{\text{target}}$, which includes English and all target languages. For each statement, we construct *contrastive* contexts where only the document to be cited, $d_{c_i}$, is presented in a language $\ell \in \mathcal{L}_{\text{eval}}$, while all other evidence documents remain in English. The full context is denoted as $\text{Context}(d_{c_i} \to \ell, d_{\neg c_i} \to \text{en})$. Given the prompt prefix $x_i$, the model's next token probability of the correct citation ID token $c_i$ corresponding to document $d_{c_i}$ conditioned on this context is: $p_\theta^{(\ell)}(c_i) = \mathcal{P}_\theta(t = c_i | x_i, q, \text{Context}(d_{c_i} \to \ell, d_{\neg c_i} \to \text{en}))$, where $\mathcal{P}$ is the model's next token distribution given a prefix, and $\theta$ denotes model parameters. We define the model's top-predicted citation token as: $\hat{c}_i^{(\ell)} = \arg\max_t(p_\theta^{(\ell)})$, and compute citation accuracy in language $\ell$ over $n$ statements as:

$$\mathbf{Acc}^{(\ell)} = \frac{1}{n} \sum_{i=1}^{n} \mathbb{1}(\hat{c}_i^{(\ell)} = c_i). \quad (2)$$

A model exhibits English preference over a target language $\ell_{\text{target}} \in \mathcal{L}_{\text{target}}$ if it achieves higher citation accuracy when the cited document $d_{c_i}$ is in English than when it is in the target language. We define the citation accuracy gap as:

$$\Delta(\ell_{\text{target}}) = \mathbf{Acc}^{(\ell_{\text{target}})} - \mathbf{Acc}^{(\text{en})}. \quad (3)$$

In other words, $\Delta(\ell_{\text{target}})$ quantifies how much more accurately the model cites English documents compared to the target language, with all other documents fixed to English. To ensure differences in raw scores are statistically meaningful, we perform pairwise two-sided $t$-tests and apply a Bonferroni correction to account for multiple comparisons.

## 4. Experiment Setup

**Dataset.** We use ELI5 dataset (Fan et al., 2019) of long-form questions from the Reddit forum "Explain Like I'm

Five." We adopt the WebGPT test set (Nakano et al., 2022) (270 queries), with relevant evidence documents for each query collected by human annotators using Bing. To successfully answer a query, the generated output must cite *all* provided relevant documents. To ensure the citation IDs are tokenized as single tokens across all evaluated models, we only use queries with $K < 10$ evidence documents. Detailed dataset statistics are in Appendix Table 2.

**Languages.** For $\mathcal{L}_{\text{target}}$, we study eight languages representing a diverse range of resource levels (measured by number of speakers and Wikipedia articles), language families, scripts, linguistic typologies: Arabic (ar), Bengali (bn), Spanish (es), French (fr), Korean (ko), Russian (ru), Swahili (sw), and Chinese (zh). Detailed characteristics per language are outlined in Appendix Table 3.

**Models.** We use six open-weight LLMs that provide full-access to model weights and support large enough context windows to handle long-context evidence documents and long-form generations. To assess the generality of language preference, we evaluate models varying in size, degree of multilinguality, and architecture family: LLaMA-3.1 8B and LLaMA-3.3 70B (Grattafiori et al., 2024), QWEN-3 8B and 14B (Yang et al., 2025), GEMMA-3 27B (Team et al., 2025), and AYA23 8B (Aryabumi et al., 2024). Details for each model can be found in Appendix Table 4.

## 5. Evidence of an English Preference

We seek to understand whether models prefer citing evidence documents in English over other languages in long-form mRAG. To do so, we analyze language preference in a controlled setup where all provided evidence documents are relevant to the query. We begin by comparing citation accuracies across languages, then explore factors that may impact language preference (§5.1). We then perform a layer-wise analysis of model behavior to unfold how language preference *evolves* (§5.2).

### 5.1. Do Models Preferentially Cite English Documents?

We define a model exhibits language preference for citing English evidence over the target language if its citation accuracy is higher for English ($\Delta(\ell_{\text{target}}) < 0$ in Eq. 3). Table 1 presents citation accuracies by model and language. Overall, we see a consistent English preference across all tested models and target languages.[7] **Even models explicitly trained on diverse languages and multilingual tasks, such as AYA23 8B, display this preference.** In Appendix D.3, we further show that, for all models, the next token probabil-

---

[7]In Appendix D.2, our embedding-similarity analysis shows that English preference cannot be fully explained by semantic similarity between the query and the cited document *alone*.

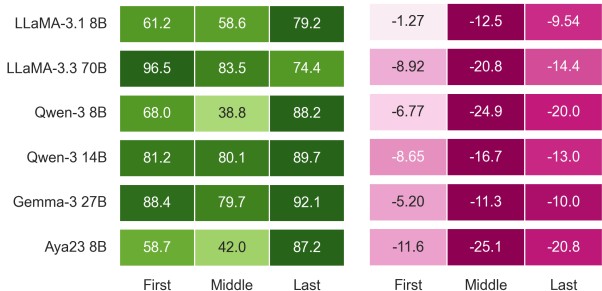

*Figure 2.* **English accuracy (*left*) and the average of $\Delta(\ell_{\text{target}})$ (*right*) (%) binned by relative position.** Each bin is normalized by sample size. $\Delta(\ell_{\text{target}})$ is largest when the cited document is positioned in the middle, indicating that position bias further amplifies English preference.

ity of the correct citation ID is the highest—and both the Shannon entropy and perplexity of the next token distribution is the lowest—when the cited document is in English, indicating models are not only more *accurate* but also more *confident* in their correct predictions for English. We also find that smaller models (8B) have lower English baseline accuracy than larger models (e.g., LLAMA-3.1 70B, GEMMA-3 27B), suggesting that models' general ability to correctly cite English evidence documents tends to improve with model scale.

**Stronger English Preference over Lower-resource Languages.** Having established an overall preference for citing English documents, we next examine which factors amplify this preference. Using the $\Delta(\ell_{\text{target}})$ values from Table 1 (i.e., the drop in citation accuracy relative to English), we find a clear correlation with language resource level: lower-resource languages exhibit largest accuracy decreases. For example, Swahili shows the greatest drop (-23.9% on average, up to -37.6% in AYA23 8B), followed by Bengali (-18.0% on average, up to -32.8% in AYA23 8B), even for models that officially support these languages (QWEN-3 8B, 14B, GEMMA-3 27B; Appendix Table 4). In contrast, higher-resource languages such as Spanish and French show smaller decreases (-8.08% and -8.82% on average, respectively), indicating weaker English preference.

**Position Bias Amplifies Language Preference.** We find that the relative position of an evidence document within the input context impacts citation accuracy. Figure 2 (*left*) shows English citation accuracy binned by the relative position of the cited document: at the beginning (First), the end (Last), or elsewhere (Middle) in the input context. Accuracy is generally lowest when the document appears in the middle (one exception is LLAMA-3 70B, which shows the lowest accuracy for the Last position). This aligns with the "lost in the middle" phenomenon, where LLMs struggle to access and use information in the middle of long contexts (Liu et al.,

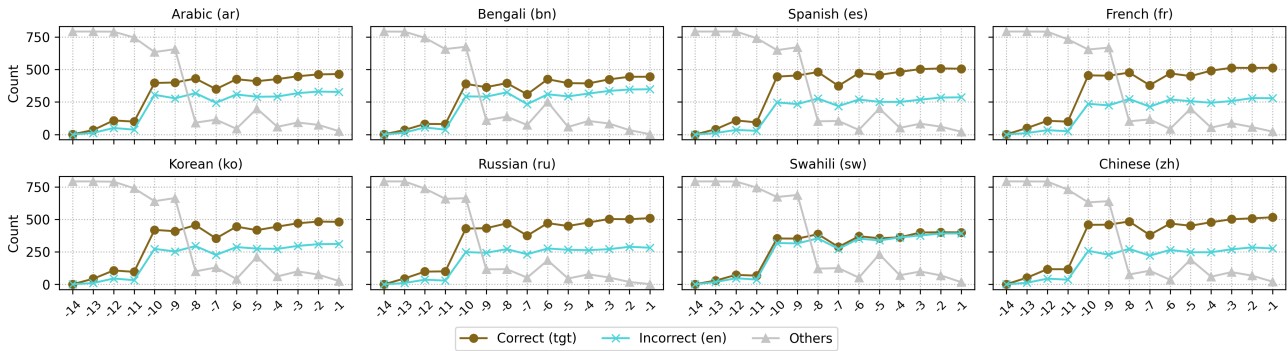

*Figure 3.* **Logit lens visualization per language for LLAMA-3.1 8B (32 layers).** $x$-axis: Last layer index; $y$-axis: Statement count. ●: Correct citation ID of document in target language; ✕: Incorrect citation ID of document in English; ▲: Not in valid citation set. Model makes a specific decision point when selecting which document to cite and largely preserves this choice across later layers. We only show last 14 layers. Results for other models are provided in Appendix D.5.

2024), here demonstrated for citation generation. Figure 2 (*right*) presents the difference in accuracy between English and the average of target languages across these positions. For all models, the largest drop in accuracy occurs when the cited document is positioned in the middle, indicating that document position not only impacts English accuracy but also amplifies models' English preference. Results for each target language are provided in Appendix D.4.

In sum, we provide strong evidence that models preferentially cite English evidence documents over target languages. This finding holds not only for *corroborative* attribution, which identifies sources that support a statement, but also for *contributive* attribution, which captures sources that causally influence the model's generation, showing consistent trends (see Appendix E). We further identify two key factors that amplify this preference: the resource level of the language and the position of the document within the input context.[8]

### 5.2. Model Layer-wise Analysis

While our earlier results confirm a strong English preference in citation, we still lack a deeper understanding on *how* this preference unfolds during generation. Does the model settle on its initial choice and persist with it or does it initially favor English documents before shifting toward the correct target language citation? This question extends prior findings from short-form tasks, where multilingual LLMs often align their internal representations with English in early layers, transitioning to target language-specific spaces only in the final layers (Wendler et al., 2024; Zhong et al., 2025; Wang et al., 2025a; Bafna et al., 2025; Schut et al., 2025). We ask whether citation generation in long-form setup follows a similar trajectory: do models initially gravi-

tate toward citing English documents and only later correct themselves, or is the outcome largely decided as soon as the model chooses which document to cite?

To probe this, we employ logit lens (Nostalgebraist, 2020), which maps intermediate layer representations into the vocabulary space, allowing us to track how token predictions evolve across layers.[9] Since logit lens is tailored to probe a single token, our citation format is a single digit, and this approach works well for this use case. For each statement, we check whether the top-1 token prediction at a given layer is (1) the correct citation ID $c_i$ (target language document; ●), (2) an incorrect ID $c_j$ ($j \neq i$, English document; ✕), or (3) not a valid citation token ($\notin \{1, \ldots, K\}$, Others; ▲).

Figure 3 shows results for LLAMA-3.1 8B. Across all languages, layers 1-17 yield no valid predictions, indicating that the model has not yet figured out the expected output format. Around layers 18-20, both correct and incorrect citation IDs begin to appear, with correct IDs slightly more frequent. Layer 22 marks a sharp peak for both correct and incorrect predictions, suggesting this is the stage where the model settles on the output format and citation predictions crystallize in the vocabulary space. From layer 23 onward, incorrect IDs remain at a stable rate, showing that once the model commits to an incorrect citation, it rarely changes. Meanwhile, count for correct IDs steadily increases, replacing the earlier invalid predictions (Others). We also find that the gap between correct and incorrect predictions narrows notably for lower-resource languages (i.e., Bengali, Swahili), confirming our earlier findings that these languages exhibit a stronger English preference.

Overall, these results indicate that models do not initially

---

[8]We further show that our findings remain robust to both—(1) stylistic variations in the citation ID (e.g., IDs expressed in different languages or formats; Appendix G) and (2) language variants in the non-cited evidence documents (Appendix F).

[9]While logit lens operates in the vocabulary space and cannot precisely localize where semantic decisions are made (Peng et al., 2025), it serves as a useful descriptive tool to identify consistent behavioral patterns in citation predictions.

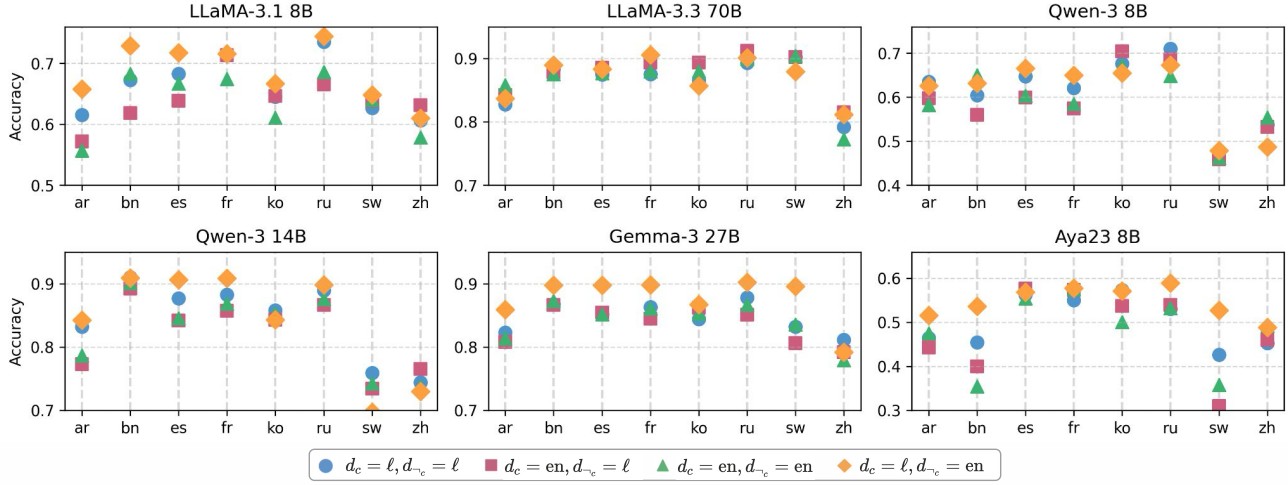

*Figure 4.* **Accuracy per model for queries in the target language.** ●: $d_c = \ell, d_{\neg c} = \ell$; ■: $d_c = \mathrm{en}, d_{\neg c} = \ell$; ▲: $d_c = \mathrm{en}, d_{\neg c} = \mathrm{en}$; ◆: $d_c = \ell, d_{\neg c} = \mathrm{en}$. Note that $y$-axis scale vary by model. $x$-axis denotes each target language. Models generally exhibit query language preference. Detailed numerical results are provided in Appendix D.6.

favor citing incorrect English documents and then switch to the correct target language. Instead, they exhibit a consistent transition region (e.g., around layers 20–22 for LLAMA-3.1 8B) where citation predictions emerge and stabilize in the vocabulary space. After this, the model largely preserves its initial prediction, whether correct or incorrect.

## 6. Effect of the Query Language

Our previous analysis demonstrate that models preferentially cite English evidence documents over those in other languages. A natural follow-up question is whether this pattern persists when the query itself is in a language other than English: do models still prefer English documents, or do they prefer documents in the same language as the query?

**Setting.** We follow the same procedure used to measure English preference (§3), with one modification in **Step 2** (**Reference Report Generation**). Each user query is translated into the target language $q_{\text{target}}$, and for each, we generate a reference citation-supported report $r_{\text{target}}$ using $K$ relevant evidence document translations $\mathcal{D}_{\text{target}}$.[10] For **Step 4** (**Next Token Prediction Analysis**), we consider four context variants differing in the language of the cited document $d_c$ and the remaining evidence documents $d_{\neg c}$: (1) Both $d_c$ and $d_{\neg c}$ in the query language ($\ell$) (●); (2) $d_c$ in English, $d_{\neg c}$ in $\ell$ (■); (3) Both $d_c$ and $d_{\neg c}$ in English (▲); (4) $d_c$ in $\ell$, $d_{\neg c}$ in English (◆). Higher citation accuracy for variants ● and ◆ compared to ■ and ▲ indicates that the model prefers citing documents in the query language. Conversely, higher accuracy for ■ and ▲ suggests a persistent English

---

[10]We use Google Translate API for query translation, with translation quality reported in Appendix Table 5.

preference regardless of the query language.

**Results.** We report citation accuracies for the four variants in Figure 4, broken down by target language for each model. Across more than half of the model-language combinations (28 out of 48), we observe the highest citation accuracy when the cited document is in the query language and all other documents are in English (◆). In 17 of these 28 cases, the second-best performance is when all documents are in the query language (●). Since the ◆ configuration generally outperforms the ● variant, this suggest that models benefit from a language contrast between the cited and the remaining documents rather than simply having more documents match the query language. French follows this trend most strongly, with 4 out of 6 models exhibiting it. One possible explanation is that, as a relatively high-resource language, models have strong enough French representations, allowing them to effectively leverage the contrast and identify the most relevant document in context.

We further see that smaller models (8B) generally achieve lower accuracies than larger models (e.g., LLAMA-3.3 70B, GEMMA-3 27B), extending our earlier observation from Section 5.1 that model size improves citation accuracy for English to non-English settings as well. Larger models also exhibit citation accuracies that are more tightly clustered across the four variants, suggesting greater robustness to language variation in the input context.

Together, these results suggest that query language plays a key role in models' language preference: models tend to favor citing documents in the same language as the query, even when that language is not English. Interestingly, this mirrors findings in scientometrics literature, where humans also exhibit an "own-language preference," tending to select

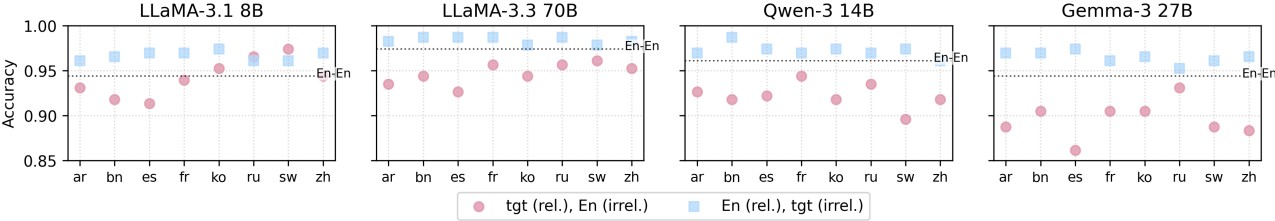

*Figure 5.* **Accuracy per model with one relevant and one irrelevant evidence document in different languages.** ●: Relevant doc in target language, irrelevant doc in English; ■: Relevant doc in English, irrelevant doc in target language; ·····: Baseline, both docs in English. Models trade off document relevance for language preference. Results for remaining models are in Appendix Figure 5 and full numerical results can be found in Appendix D.7.

and cite sources in the language of their writing (Yitzhaki, 1998; Egghe et al., 1999).

## 7. Relevance vs. Language Preference

Sections 5 and 6 analyzed language preference in a controlled setup where all provided evidence documents were relevant to the query. In reality, however, retrievers are *imperfect*, and retrieved evidence often contains irrelevant or partially relevant documents (Chen et al., 2024a; Jin et al., 2025). To better approximate such conditions, we relax the assumption that all documents are relevant and ask: between relevance and language, which exerts a stronger influence on model citation behavior?

**Setting.** We compare the effects of document relevance and language by varying the language of one relevant and one irrelevant document under three conditions: (1) **En-En**: Both relevant and irrelevant documents are in English; (2) **tgt-En** (●): Relevant document in the target language, irrelevant document in English; (3) **En-tgt** (■): Relevant document in English, irrelevant document in the target language. We constrain to this setup to ensure that the model's decision can vary only along these two dimensions.

Since ELI5 dataset does not include irrelevant documents, we use MIRACL (Zhang et al., 2023), a multilingual RAG dataset with Wikipedia queries. MIRACL simulates a realistic retrieval setting as the irrelevant documents are collected via (1) retrieving candidate passages from the query and a Wikipedia dump, and (2) selecting those labeled "irrelevant" by human annotators. Therefore, they are often topically related to the query but not necessary for answering it, simulating realistic retrieval noise. We use the English subset of the development set, restricting to queries with exactly one relevant document (231 queries). We randomly use one of the irrelevant documents. For each query, we follow the same process described in Section 3.

**Results.** Our hypotheses are: (*i*) If citation accuracy in **tgt-En** is lower than the **En-En** baseline, it suggests the

model is overly influenced by language, preferring to cite an irrelevant English document over a relevant target language one, and (*ii*) if citation accuracy in **En-tgt** exceeds the baseline, it implies that the model more easily ignores irrelevant target language distractors, again signaling English preference.

Our results support both hypotheses (Figure 5). When the relevant document is in the target language, accuracies consistently drop below the baseline, indicating that irrelevant English content more easily mislead the model. We show qualitative example in Appendix H. Conversely, accuracies for all languages and models rise above the baseline for **En-tgt**, suggesting that target language distractors are easier to dismiss than English distractors. This aligns with recent findings that distractors in the same language as the relevant document degrade performance more severely (Qi et al., 2025). One interesting observation is Swahili. Although it yields the lowest accuracies in the ELI5 experiments (see Table 1), its performance in the **En-tgt** setup is relatively strong. One possible explanation is its use of the Latin script, shared with English, which may make irrelevant Swahili documents appear more plausible.[11]

## 8. Conclusion

We propose a controlled methodology to measure language preference in long-form mRAG by isolating language effects while controlling for document content and relevance. Our analysis shows that models preferentially cite English documents when queries are in English, with this bias stronger for lower-resource languages and mid-context. Importantly, this preference can outweigh relevance, with models often citing irrelevant English documents over relevant non-English ones. Overall, our findings demonstrate how model internals reveal citation behavior in mRAG and offer insights for designing more robust, inclusive systems that balance language and relevance.

---

[11]We further show that models also trade-off relevance for language preference when queries are posed in a language other than English in Appendix I.

## Limitations

The dataset used in our main experiments, ELI5 (Fan et al., 2019), has known limitations—such as substantial train-validation overlap and answers that are not often grounded in the supporting documents (Krishna et al., 2021). However, ELI5 was the *only* publicly available dataset that met the requirements of our setup. To complement this, we additionally run experiments on MIRACL (Zhang et al., 2023) in Appendix K, and observe the same English (§5) and query language preference (§6).

Our analysis uses a controlled setup with several simplifying assumptions: (1) retrieval is complete and all evidence documents are equally relevant and (2) multilingual RAG is simulated via MT of English documents since no parallel long-form mRAG datasets are publicly available. To justify our use of MT, we show that (a) English preference does not meaningfully correlate with MT quality (Appendix J), (b) our findings remain consistent when using naturally occurring queries in the target language (Appendix K.2), and (c) with an alternative MT system (Appendix L). These assumptions may not fully hold in real-world settings, which could limit the generalizability of our results. We provide complementary experiments simulating an end-to-end mRAG framework in Appendix M, where all metrics are consistently higher when all documents are in English—corroborating the English preference observed using by our citation accuracy metric. Nonetheless, our study provides valuable insights into language preference that can guide future work on understanding and improving model citation behavior.

## Impact Statement

Multilingual RAG systems are increasingly used to support information access and decision-making across languages and cultures. Our findings show that such systems exhibit systematic language-based citation preferences, particularly favoring English documents even when they are not the most relevant. If left unexamined, these biases risk marginalizing non-English sources, reinforcing existing language hierarchies, and reducing trust in AI-assisted tools for multilingual users. By identifying where and how these preferences arise within the model, this work highlights the importance of more transparent and language-aware evaluation practices. We hope this analysis informs the development of more equitable multilingual systems and encourages practitioners to critically assess citation behavior in high-stakes applications such as healthcare, education, and journalism.

## Acknowledgements

We would like to thank the members of the Johns Hopkins University SCALE 2025 program. We are grateful to the generation team who gave constructive feedback and support in shaping this work. Dayeon also extends special thanks to the friends for making the internship experience in Baltimore truly memorable, including Yu Hou, Bryan Li, Gabrielle Kaili-May Liu, Maxime Dassen, Roxana Petcu, Jia-Huei Ju, Francois Landry, and Siddharth Singh. This work was supported in part by NSF Fairness in AI Grant 2147292, and by the Institute for Trustworthy AI in Law and Society (TRAILS), which is supported by the National Science Foundation under Award No. 2229885. The views and conclusions contained herein are those of the authors and should not be interpreted as necessarily representing the official policies, either expressed or implied, of NSF or the U.S. Government. The U.S. Government is authorized to reproduce and distribute reprints for governmental purposes notwithstanding any copyright annotation therein.

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

# Appendix

## A. Prompts

We present the prompts used for generating the gold citation-supported report (Figure 6), obtaining supportedness judgments from LLM-as-judge (Figure 7), and guessing the next token predictions from the evaluated models (Figure 8). We adopt base prompts from GPTResearcher (Elovic, 2023).

---

**Prompt A.1. Gold Report Generation Prompt**

**Information:**
**Document ID:** {document ID}
**Title:** {title}
**Content:** {content}
—

...

—
Using the above information, respond to the following query or task: {query}.
The response should focus on the answer to the query, should be well structured, informative, and concise, with facts and numbers if available.

Please follow all of the following guidelines in your response:
- You MUST write in a single paragraph and at most {total words} words.
- You MUST write the response in the following language: {language}.
- You MUST cite your sources, especially for relevant sentences that answer the question.
- When using information that comes from the documents, use citation which refer to the Document ID at the end of the sentence (e.g., [1]).
- Do NOT cite multiple documents at the end of the sentence (e.g., [1][2]).
- If multiple documents support the sentence, only cite the most relevant document.
- It is important to ensure that the Document ID is a valid string from the information above and that the information in the sentence is present in the document.

**Response:**

---

*Figure 6.* **Prompt for generating gold citation-supported reports.** Information section is populated with the document ID, title, and content of each evidence document. Boldface is only for emphasis.

---

**Prompt A.2. LLM-as-judge Prompt**

**Instruction:** You are given a query, a document, and a sentence from a generated response that cites the document in answering the query. Determine which document best supports the information in the cited sentence. Respond only with the exact document ID. Do not provide any additional explanation.

**Query:** {query}
**Information:**
**Document ID:** {document ID}
**Title:** {title}
**Content:** {content}
—

...

—

**Cited sentence:** {statement}
**Response:**

---

*Figure 7.* **Prompt for getting supportedness judgments from LLM-as-judge.** Information section is populated with the document ID, title, and content of each evidence document. Boldface is only for emphasis.

---

**Prompt A.3. Next Token Prediction Prompt**

**Information:**
**Document ID:** {document ID}
**Title:** {title}
**Content:** {content}
—
...
—
Using the above information, the response is the answer to the query or task: {query} in a single sentence.
You MUST cite the most relevant document by including only its Document ID in brackets at the end of the sentence (e.g., [Document ID]).
Do NOT include any additional words inside or outside the brackets.
Please output ONLY the number of the Document ID that is most relevant to the sentence.

**Response:** {statement}[

---

*Figure 8.* **Prompt for guessing the next token prediction.** Information section is populated with the document ID, title, and content of each evidence document. Boldface is only for emphasis.

## B. Details of Dataset, Languages, and Models

We provide detailed statistics of the two long-form RAG datasets used in our experiments (ELI5 and MIRACL) in Table 2. The characteristics of the eight tested languages, including their language family, script, linguistic typology, and resource level, are summarized in Table 3. For the models, Table 4 includes their context window size, HuggingFace model identifier, and officially (un)supported languages. Lastly, Table 5 reports COMET-QE (Rei et al., 2020) scores for each target language.

| Dataset | # Queries | Avg. # Words ($q$) | Avg. # Words ($t$) | Avg. # Words ($d$) | Avg. # Sent ($d$) | Avg. # $d$ per $q$ |
|---------|-----------|--------------------|--------------------|--------------------|--------------------|---------------------|
| **ELI5** | 270 | 15.25 | 9.64 | 76.82 | 4.26 | 3.49 |
| **MIRACL** | 231 | 6.87 | 2.63 / 2.83 | 106.59 / 115.80 | 5.41 / 5.88 | 1.00 / 9.31 |

*Table 2.* **Detailed statistics of long-form RAG datasets used.** We report statistics for ELI5 (Explain Like I'm Five) and MIRACL. For MIRACL, statistics are shown as relevant / irrelevant documents. $q$: query; $t$: title, $d$: evidence document.

| Language Family | Language | Script | Synthesis | Word Order | Resource Level | # Speakers | # Wikipedia Size |
|-----------------|----------|--------|-----------|------------|----------------|------------|------------------|
| Indo-European | English | Latin | analytic | SVO | high | 1,130M | 5,758,285 |
| | French | Latin | fusional | SVO | high | 398M | 2,325,608 |
| | Spanish | Latin | fusional | SVO | high | 592M | 1,669,181 |
| | Russian | Cyrillic | fusional | SVO | mid | 260M | 1,476,045 |
| | Bengali | Bengali | fusional | SOV | low | 337M | 63,762 |
| Sino-Tibetan | Chinese | Chinese | analytic | SVO | high | 1,350M | 1,246,389 |
| Koreanic | Korean | Hangul | agglutinative | SOV | mid | 128M | 1,133,444 |
| Afro-Asiatic | Arabic | Arabic | fusional | VSO | mid | 630M | 656,982 |
| Niger-Congo | Swahili | Latin | agglutinative | SVO | low | 83M | 47,793 |

*Table 3.* **Characteristics of tested languages.** For each language, we show language family, script, linguistic typologies (synthesis and word order), and resource level measured by the number of speakers and Wikipedia articles (Zhang et al., 2023).

| Model | Context Window | HuggingFace Model Identifier | Supported Langs | Unsupported Langs |
|---|---|---|---|---|
| **LLAMA-3 8B** | 128K | `meta-llama/Llama-3.1-8B-Instruct` | en, es, fr | ar, bn, ru, ko, sw, zh |
| **LLAMA-3 70B** | 128K | `meta-llama/Llama-3.3-70B-Instruct` | en, es, fr | ar, bn, ru, ko, sw, zh |
| **QWEN-3 8B** | 33K | `Qwen/Qwen3-8B` | en, ar, bn, es, fr, ru, ko, sw, zh | - |
| **QWEN-3 14B** | 33K | `Qwen/Qwen3-14B` | en, ar, bn, es, fr, ru, ko, sw, zh | - |
| **GEMMA-2 27B** | 128K | `google/gemma-3-27b-it` | en, ar, bn, es, fr, ru, ko, sw, zh | - |
| **AYA23 8B** | 8,192 | `CohereLabs/aya-23-8B` | en, ar, es, fr, ru, ko, zh | bn, sw |

*Table 4.* **List of evaluated models.** We report the context window size, HuggingFace model identifiers, and the *officially* supported languages during pretraining. Note: Supported language information is extracted from each model's technical report. We use ISO 639-1 codes for languages. We use QWEN-3 series models with `enable_thinking=False` mode.

| Language | COMET-QE($q$, $q'$) | COMET-QE($t$, $t'$) | COMET-QE($d$, $d'$) |
|---|---|---|---|
| **Arabic** | 0.752 | 0.541 | 0.511 |
| **Bengali** | 0.824 | 0.584 | 0.559 |
| **Spanish** | 0.823 | 0.583 | 0.564 |
| **French** | 0.822 | 0.582 | 0.566 |
| **Korean** | 0.816 | 0.584 | 0.555 |
| **Russian** | 0.780 | 0.557 | 0.528 |
| **Swahili** | 0.769 | 0.544 | 0.516 |
| **Chinese** | 0.777 | 0.561 | 0.534 |

*Table 5.* **COMET-QE scores by language.** We evaluate the machine translation (MT) quality of non-English queries ($q$), titles ($t$), and evidence documents ($d$) in the ELI5 dataset. Apostrophe ($'$) indicates MT. Higher scores indicate better MT quality.

# C. Justification for NLI Filtering

## C.1. Human Annotation

To validate the two-step automatic filtering process described in Section 3 for identifying supported statements, we conduct a small-scale human annotation study on 60 sampled statements. We stratify the sample into 30 "supported" statements (passing both the LLM-as-Judge and NLI entailment filters and included in the final statement pool) and 30 "unsupported" statements (failing one or both filters). We conducted a power analysis to justify our sample size. Using a $t$-test for 2 independent samples[12], we find that 26 statements per label group (supported and unsupported, total 52) are required to detect a minimum effect size of Cohen's $d$ of 0.8 with a significance level of $\alpha$ of 0.05, and desired power of 0.8.

For each query $q$, statement $s_i$, and cited document $d_{c_i}$, we ask annotators: "How well is the statement supported by the provided document?" Responses are given on a five-point Likert scale from 5 (Definitely) to 1 (Not at all), using instructions similar to those provided when prompting the judge LLMs (Figure 7). Figure 9 shows the full instructions and an example provided to annotators.

We recruit six annotators from Prolific[13] who resides in the United States with first, primary, and fluent language as English. We compensate each with USD 8 (equivalent to USD 16/hour), totaling USD 56 including Prolific platform fees. Each annotator evaluates 30 statements (15 supported and 15 unsupported) presented in randomized order. Inter-annotator agreement is moderate, with a Krippendorff's alpha of 0.559. The average rating for supported statements is 4.15 out of 5, while unsupported statements average 2.49 out of 5. These results indicate strong alignment between our automatic filtering process and human judgments of statement supportedness. Figure 10 plots the rating distribution for each label group.

## C.2. End-to-end Evaluation

We implement an end-to-end evaluator inspired by RAGAS (Es et al., 2024) that uses a single LLM-as-judge (OPENAI O3) to rate each statement for (1) context relevance (which document best supports it) and (2) faithfulness (whether the statement is faithful to the cited document), retaining only statements that pass both. We compare this to our original pipeline

---

[12]https://www.statsmodels.org/stable/generated/statsmodels.stats.power.TTestIndPower.html
[13]https://www.prolific.com/

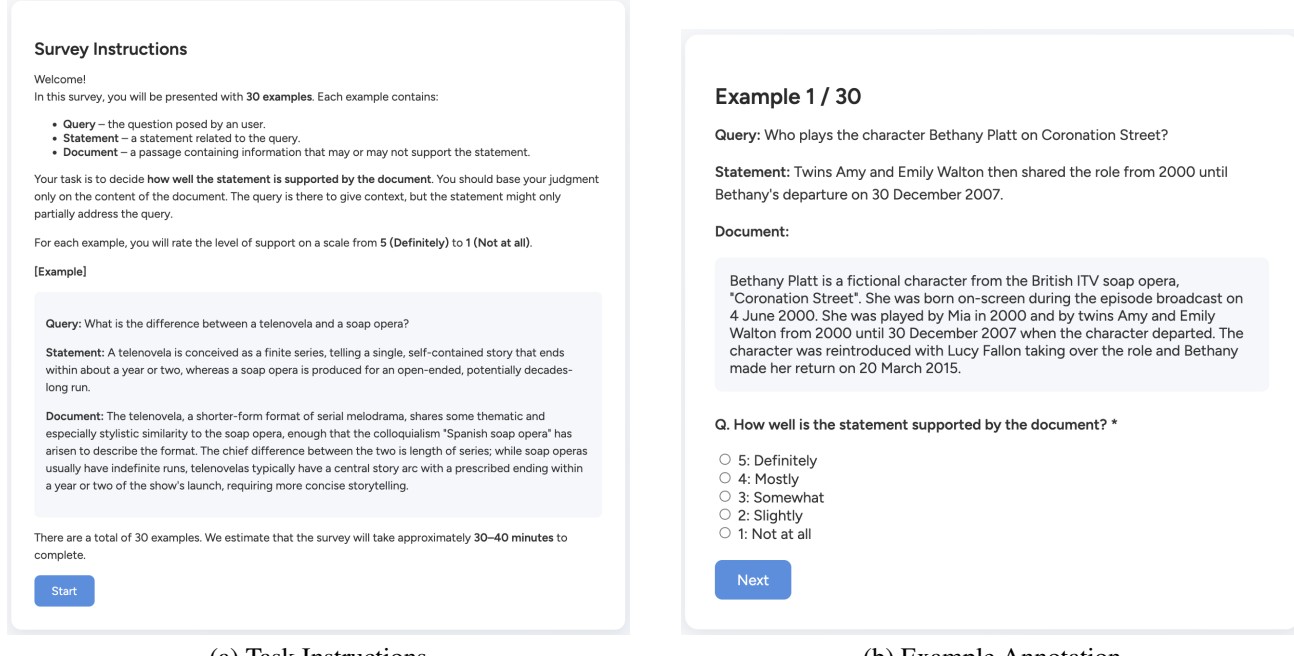

(a) Task Instructions                                    (b) Example Annotation

*Figure 9.* **Full instructions and example provided to human annotators.** The annotation task was hosted on a custom-built website. Annotators first viewed a brief task instruction (a), then evaluated 30 statements, with an example shown in (b).

(LLM-relevance-judge + NLI) in terms of precision (% of statements retained by the end-to-end evaluator, also retained by our original method), recall (% of statements retained by our method, also retained by the end-to-end evaluator), and F1 (harmonic mean of precision and recall). In Table 6, we observe a substantial overlap between the two evaluators across languages, with an overall F1 of 0.847. This supports that our NLI-based verification is robust and well aligned with an end-to-end RAGAS-style evaluator.

| Language | Precision | Recall | F1 |
|---|---|---|---|
| **Arabic** | 0.774 | 0.874 | 0.821 |
| **Bengali** | 0.843 | 0.855 | 0.849 |
| **English** | 0.836 | 0.845 | 0.840 |
| **Spanish** | 0.843 | 0.906 | 0.873 |
| **French** | 0.847 | 0.900 | 0.873 |
| **Korean** | 0.801 | 0.860 | 0.829 |
| **Russian** | 0.821 | 0.888 | 0.853 |
| **Swahili** | 0.802 | 0.892 | 0.832 |
| **Chinese** | 0.869 | 0.855 | 0.850 |
| **Overall** | 0.826 | 0.875 | 0.847 |

*Table 6.* **Comparison of our method and end-to-end evaluator.**

## D. Detailed Results

### D.1. Machine Translation Quality

We evaluate Machine Translation (MT) quality for translated queries, titles, and evidence documents using COMET-QE scores. We do not perform any filtering based on these scores. Table 5 reports average scores by language, and Figure 11 shows full score distributions. We find little evidence that MT quality drives English preference. Document COMET-QE scores (last column of Table 5) are lowest for Arabic (0.511) and Swahili (0.516), while Bengali shows a relatively high

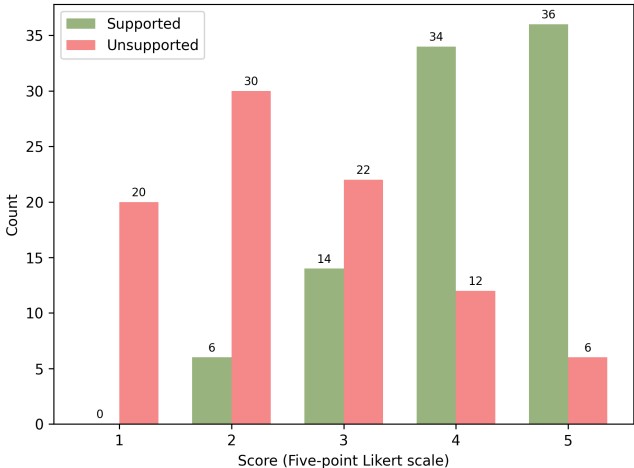

*Figure 10.* **Rating distribution for each label group.** We plot the distribution of 180 judgments collected during human annotation (90 supported and 90 unsupported statements). Results show that annotators can reliably distinguish supported from unsupported statements based on their ratings.

score (0.559). Yet, citation accuracies (Table 1) show that Arabic's ranking varies widely across models—third lowest for LLaMA-3.1 8B and Qwen-3 8B, lowest for LLaMA-3.3 70B, fourth lowest for Qwen-3 14B and Aya23, but relatively higher for Gemma-3 27B. By contrast, Bengali exhibits the second-strongest English preference after Swahili despite its higher MT quality. This suggests that resource level, rather than MT quality, is a stronger indicator of English preference.

| Language | $q = \text{en}, d_c = \text{en}$ | $q = \text{en}, d_c = \ell$ | $q = \ell, d_c = \text{en}$ | $q = \ell, d_c = \ell$ |
|---|---|---|---|---|
| **Arabic** | 0.579 | 0.584 | 0.601 | 0.653 |
| **Bengali** | 0.579 | 0.586 | 0.571 | 0.637 |
| **Spanish** | 0.579 | 0.586 | 0.575 | 0.645 |
| **French** | 0.579 | 0.589 | 0.574 | 0.648 |
| **Korean** | 0.579 | 0.577 | 0.573 | 0.654 |
| **Russian** | 0.579 | 0.589 | 0.571 | 0.654 |
| **Swahili** | 0.579 | 0.588 | 0.574 | 0.648 |
| **Chinese** | 0.579 | 0.586 | 0.573 | 0.651 |

*Table 7.* **Embedding similarity between query and cited document.** $q$: query; $d_c$: cited document, $\ell$: target language.

### D.2. Embedding Similarity Analysis

We compute embedding similarity between the query ($q$) and the cited document ($d_c$) using the multilingual encoder LaBSE (Feng et al., 2022). As shown in Table 7, when the query is in English ($q = \text{en}$), the embedding similarities show no statistically significant difference between cases where the cited document is in English vs. non-English languages ($\ell$) (columns 1-2). When the query is in a non-English language ($q = \ell$), we do observe higher similarity scores for cited documents in the same language as the query (columns 3-4). This suggests that English preference observed in Table 1 cannot be fully explained by semantic similarity alone.

### D.3. Evidence of English Preference

While our main accuracy metric is an intuitive measure, more fine-grained probability changes might not be captured. Therefore, for each model and language, we report the next token probability assigned to the correct citation ID (Table 8) and the Shannon entropy of the next token distribution (Table 9). Across all models, we observe consistently higher probabilities when the cited evidence document is in English, alongside lower entropy values.

We further report the perplexity values in Table 10. We show that they are the lowest for English across all models except

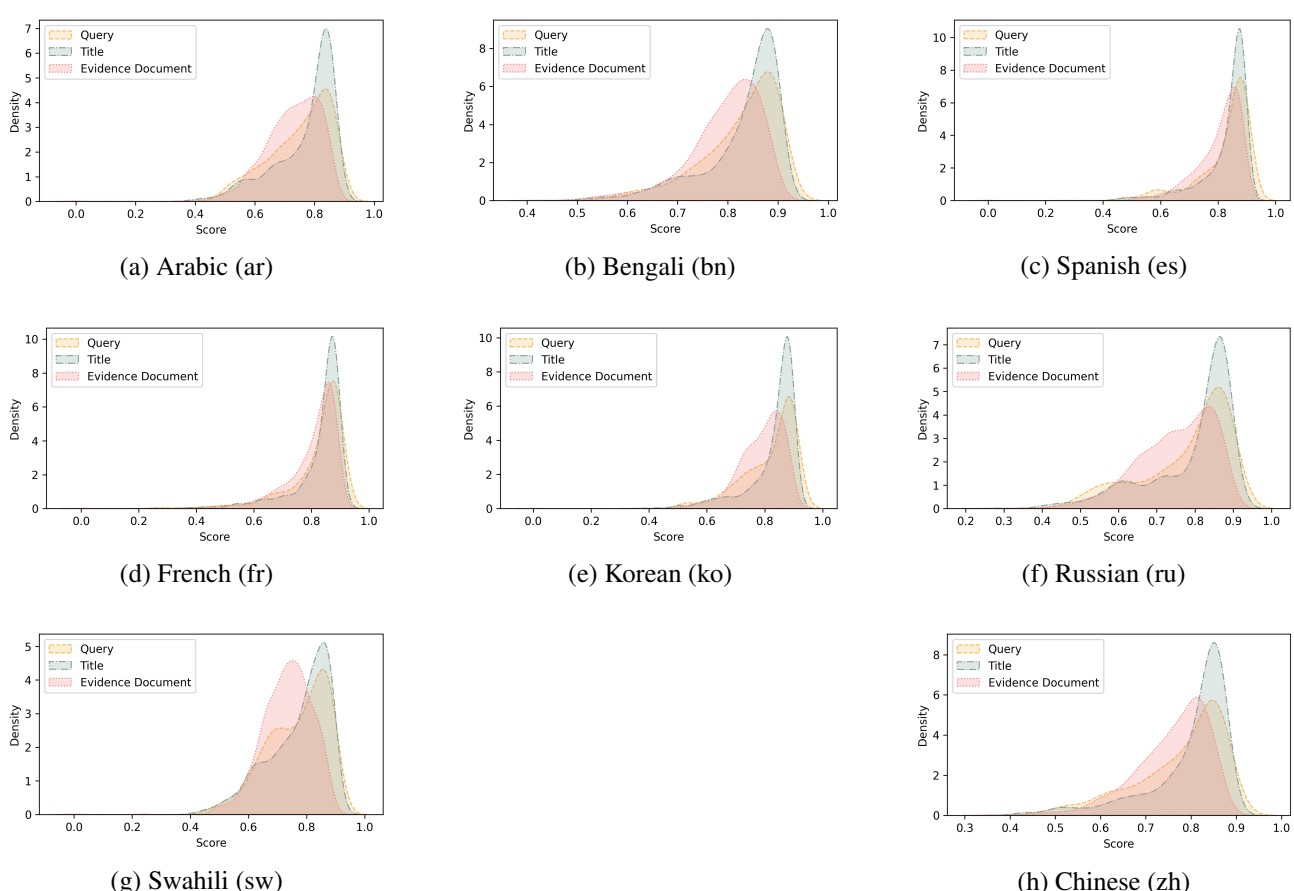

*Figure 11.* **COMET-QE score distributions by language.** Distributions are more skewed for shorter content (*e.g.,* title), while broader distributions appear for longer content (*e.g.,* evidence document).

QWEN-3 8B, where Chinese is slightly lower, but the difference is not statistically significant. Together, this suggests that models are not only more accurate but also more confident when correctly citing English documents.

| Language | LLAMA-3.1 8B | LLAMA-3.3 70B | QWEN-3 8B | QWEN-3 14B | GEMMA-3 27B | AYA23 8B |
|---|---|---|---|---|---|---|
| **English** | 0.651 | 0.991 | 0.758 | 0.984 | 0.980 | 0.527 |
| **Arabic** | 0.629 (-0.022) | 0.990 (-0.001) | 0.751 (-0.007) | 0.979 (-0.005) | 0.968 (-0.012) | 0.463 (-0.064) |
| **Bengali** | 0.647 (-0.004) | 0.990 (-0.001) | 0.736 (-0.022) | 0.981 (-0.003) | 0.977 (-0.003) | 0.442 (-0.085) |
| **Spanish** | 0.626 (-0.025) | 0.987 (-0.004) | 0.752 (-0.006) | 0.981 (-0.003) | 0.979 (-0.001) | 0.483 (-0.044) |
| **French** | 0.649 (-0.002) | 0.991 (0.000) | 0.728 (-0.030) | 0.983 (-0.001) | 0.973 (-0.007) | 0.499 (-0.028) |
| **Korean** | 0.620 (-0.031) | 0.982 (-0.009) | 0.730 (-0.028) | 0.983 (-0.001) | 0.955 (-0.025) | 0.494 (-0.033) |
| **Russian** | 0.634 (-0.017) | 0.990 (-0.001) | 0.707 (-0.051) | 0.982 (-0.002) | 0.961 (-0.019) | 0.465 (-0.062) |
| **Swahili** | 0.630 (-0.021) | 0.987 (-0.004) | 0.634 (-0.124) | 0.967 (-0.017) | 0.966 (-0.014) | 0.479 (-0.048) |
| **Chinese** | 0.642 (-0.009) | 0.988 (-0.003) | 0.706 (-0.052) | 0.984 (0.000) | 0.976 (-0.004) | 0.488 (-0.039) |

*Table 8.* **Next token probabilities for the correct citation ID by model and language (↑).** We present mean values along with the difference from English baseline indicated in subscript.

| Language | LLAMA-3.1 8B | LLAMA-3.3 70B | QWEN-3 8B | QWEN-3 14B | GEMMA-3 27B | AYA23 8B |
|---|---|---|---|---|---|---|
| **English** | 1.106 | 0.132 | 0.388 | 0.064 | 0.028 | 1.215 |
| **Arabic** | 1.146 (+0.040) | 0.176 (+0.044) | 0.500 (+0.112) | 0.088 (+0.024) | 0.063 (+0.035) | 1.277 (+0.062) |
| **Bengali** | 1.169 (+0.063) | 0.178 (+0.046) | 0.457 (+0.069) | 0.095 (+0.031) | 0.051 (+0.023) | 1.350 (+0.135) |
| **Spanish** | 1.152 (+0.046) | 0.150 (+0.018) | 0.460 (+0.072) | 0.081 (+0.017) | 0.048 (+0.020) | 1.260 (+0.045) |
| **French** | 1.122 (+0.016) | 0.149 (+0.017) | 0.389 (+0.001) | 0.075 (+0.011) | 0.051 (+0.023) | 1.247 (+0.032) |
| **Korean** | 1.150 (+0.044) | 0.166 (+0.034) | 0.394 (+0.006) | 0.087 (+0.023) | 0.059 (+0.031) | 1.269 (+0.054) |
| **Russian** | 1.134 (+0.028) | 0.162 (+0.030) | 0.412 (+0.024) | 0.074 (+0.010) | 0.059 (+0.031) | 1.266 (+0.051) |
| **Swahili** | 1.194 (+0.088) | 0.182 (+0.050) | 0.508 (+0.120) | 0.123 (+0.059) | 0.054 (+0.026) | 1.254 (+0.039) |
| **Chinese** | 1.130 (+0.024) | 0.159 (+0.027) | 0.385 (+-0.003) | 0.084 (+0.020) | 0.067 (+0.039) | 1.255 (+0.040) |

*Table 9.* **Shannon entropy by model and language (↓).** We present mean values along with the difference from English baseline indicated in subscript.

### D.4. Position-wise Accuracy per Language

We show accuracy gap between English and each target language in Figure 12. We show that the findings with the aggregated results in Section 5.1 are consistent for all languages: the accuracy drop is generally most pronounced when the cited document appears in the middle of the input context.

### D.5. Logit Lens Analysis

Figures 13 to 17 present logit lens visualizations for each model. We observe different trends:

**LLAMA-3.3 70B.** The model follows a trajectory similar to LLAMA-3.1 8B. Both the correct and wrong citation ID predictions begin to rise around layer 40, peak sharply at layers 52-57, then decline until layer 60 before increasing again and stabilizing toward the final layers. Throughout, correct predictions consistently outnumber incorrect ones. As with LLAMA-3.1 8B, the gap between correct and incorrect predictions narrows for lower-resource languages.

**QWEN-3 8B.** The model exhibits a staggered pattern, where correct citation IDs peak around layer 26, again at layers 28-30, and once more at the final layer, remaining low in between. While the model already predicts the correct IDs in earlier layers (28-30), they are overtaken by invalid predictions just before the final two layers, after which the model uncovers and ends with a final peak in accuracy.

**QWEN-3 14B.** Despite belonging to the same QWEN-3 family, this model exhibits a completely different behavior from QWEN-3 8B. For most of its layers, it fails to predict outputs in the expected citation format. Only in the final layers (38-40), we observe an increase in correct citation predictions, consistently outpacing incorrect ones. This suggests a more conservative prediction strategy, where it delays citation prediction until the very end, or it can only recognize the citation

| Language | LLAMA-3.1 8B | LLAMA-3.3 70B | QWEN-3 8B | QWEN-3 14B | GEMMA-3 27B | AYA23 8B |
|---|---|---|---|---|---|---|
| **English** | 3.023 | 1.141 | 1.474 | 1.066 | 1.029 | 3.370 |
| **Arabic** | 3.147** | 1.193*** | 1.649*** | 1.092** | 1.065*** | 3.585*** |
| **Bengali** | 3.219* | 1.194*** | 1.579*** | 1.100*** | 1.052*** | 3.857*** |
| **Spanish** | 3.164*** | 1.162** | 1.584*** | 1.085** | 1.050*** | 3.526*** |
| **French** | 3.072 | 1.161** | 1.476 | 1.078 | 1.052*** | 3.481*** |
| **Korean** | 3.159** | 1.180*** | 1.483*** | 1.091** | 1.061*** | 3.556*** |
| **Russian** | 3.109* | 1.176*** | 1.51 | 1.077 | 1.061*** | 3.548*** |
| **Swahili** | 3.300*** | 1.200*** | 1.662*** | 1.131*** | 1.055*** | 3.506*** |
| **Chinese** | 3.097 | 1.172*** | 1.47 | 1.088** | 1.070*** | 3.509*** |

*Table 10.* **Perplexity values by model and language ($\downarrow$).** We present mean values along with the difference from English baseline indicated in subscript. Pairwise two-sided $t$-tests are performed to compare perplexity between English and the target language, with the null hypothesis that the mean perplexity is equal across languages. Bonferroni correction is applied for multiple comparisons. *: significant with $p < 0.05$; **: $p < 0.01$; ***: $p < 0.001$; non-marked: not statistically significant.

format at the final layers.

**GEMMA-3 27B.** Similar to the QWEN-3 8B, this model shows a staggered pattern, where incorrect predictions remain low, while correct predictions generally increase. There are sharp drops around layers 53-54 and layer 58. However, the model recovers by the final layer, and the count of correct predictions stays high.

**AYA23 8B.** This model stands out from the others, as incorrect predictions generally outnumber correct ones. This aligns with the results in Table 1, where AYA23 8B shows the largest average accuracy drop for target languages. It is also especially pronounced for lower-resource languages like Bengali or Swahili, where the gap between correct and incorrect predictions is even wider.

### D.6. Query Language Variants

In Table 11, we report the full numerical results when the query is posed in a target language. We consider four variants, differing in the language of the cited document and the remaining evidence documents, following the same notation introduced in Figure 4: (1) $d_c = \ell, d_{\neg c} = \ell$: all documents in the query language (●); (2) $d_c = \text{en}, d_{\neg c} = \ell$: cited document in English and all other documents in the query language (■); (3) $d_c = \text{en}, d_{\neg c} = \text{en}$: all documents in English (▲); and (4) $d_c = \ell, d_{\neg c} = \text{en}$: cited document in the query language and all other documents in English (◆). Overall, we find that models tend to prefer citing evidence in the query language, with $d_c = \ell, d_{\neg c} = \text{en}$ configuration (◆) achieving the highest accuracy in more than half of the cases.

### D.7. Relevance vs. Language Preference

In Figure 18, we plot citation accuracy for the remaining models (QWEN-3 8B and AYA23 8B) with one relevant and one irrelevant evidence document in different languages, complementing the results in Figure 5. Table 12 reports the full numerical results using the notation from Section 7: (1) **En-En**: both relevant and irrelevant documents are in English, (2) **tgt-En**: relevant document in the target language and irrelevant document in English, and (3) **En-tgt**: relevant document in English and irrelevant document in the target language. Overall, we observe that citation accuracy in **tgt-En** is generally lower than the **En-En** baseline, while **En-tgt** is consistently higher, both indicating a strong English preference that persists regardless of differences in document relevance.

## E. Contributive Attribution Patterns

Our analysis of language preference has been based on *corroborative* attribution, measuring the probability of generating in-line citations, which identifies sources that *support* a statement (Menick et al., 2022; Liu et al., 2023). However, if models are citing more English documents, that does not necessarily mean they are actually attributing on their content. If models truly favor English sources, we would expect that preference to also appear when we examine *contributive* attribution, which identifies sources that *cause* a model to generate a specific statement.

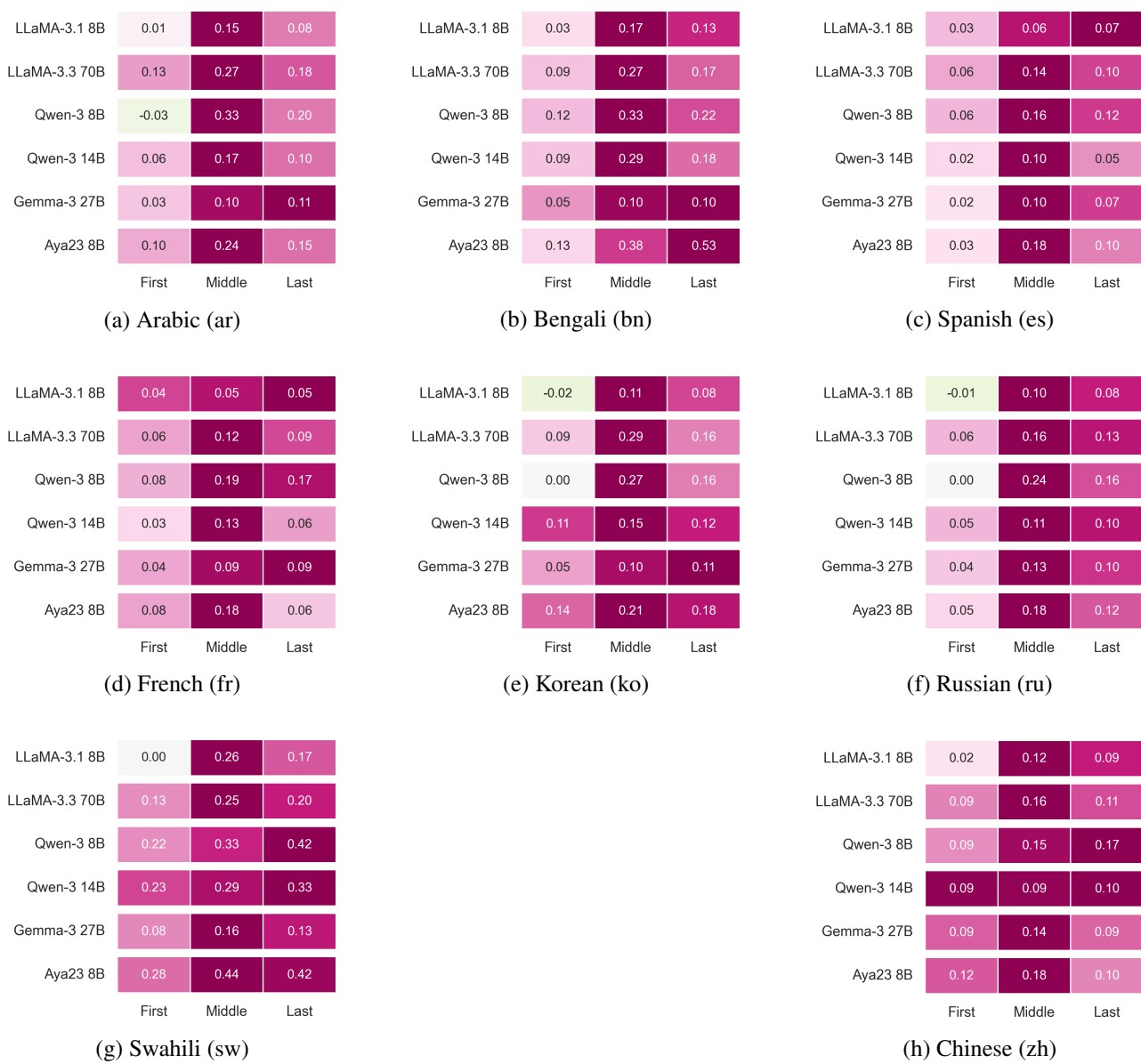

*Figure 12.* **Accuracy difference between English and each target language binned by relative position.** Each bin is normalized by sample size.

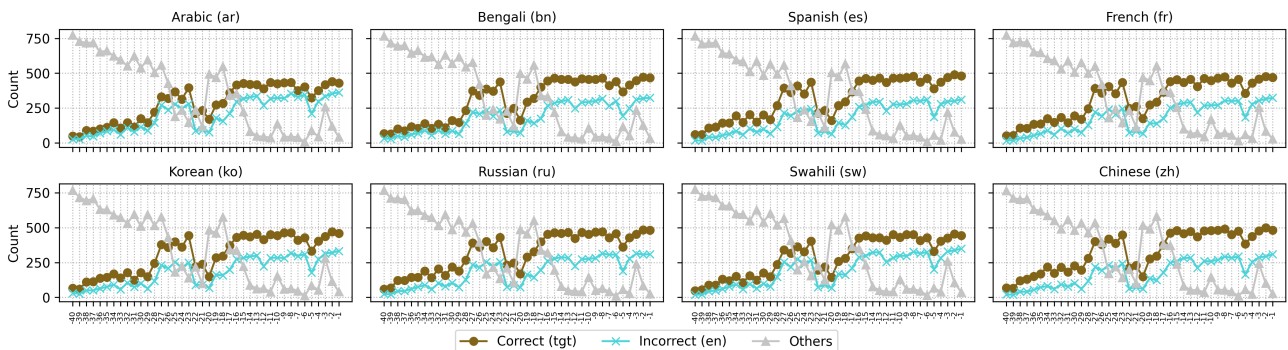

*Figure 13.* **Logit lens visualization per language for LLAMA-3.3 70B (80 layers).** $x$-axis: Last layer index; $y$-axis: Statement count. We show the last 40 layers. ●: Correct citation ID of document in target language; ✕: Wrong citation ID of document in English; ▲: Not in valid citation set.

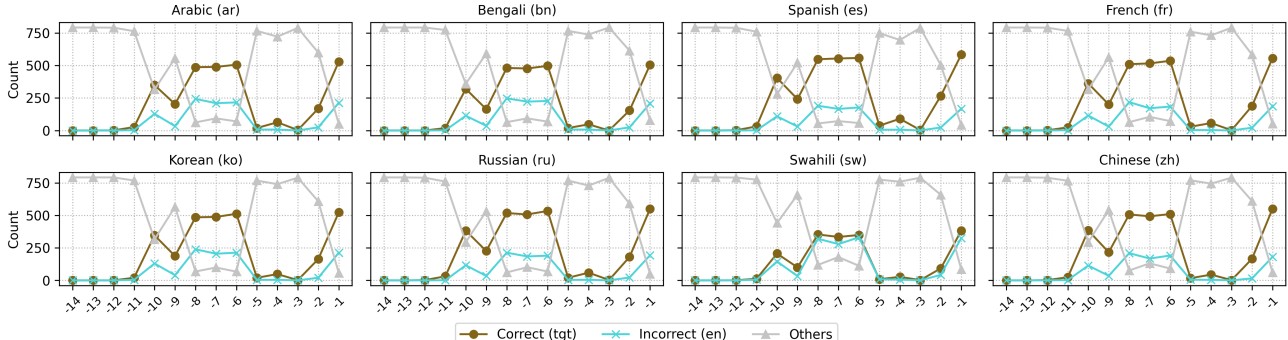

*Figure 14.* **Logit lens visualization per language for QWEN-3 8B (36 layers).** $x$-axis: Last layer index; $y$-axis: Statement count. We show the last 14 layers.

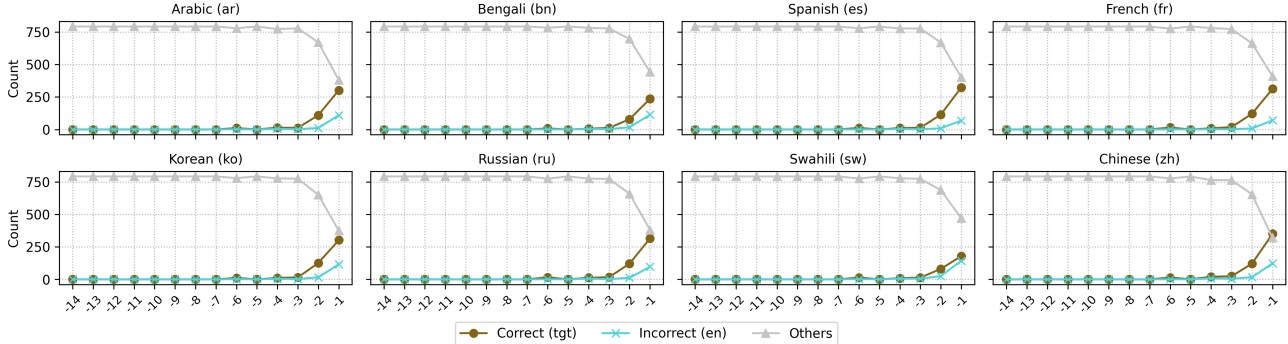

*Figure 15.* **Logit lens visualization per language for QWEN-3 14B (40 layers).** $x$-axis: Last layer index; $y$-axis: Statement count. We show the last 14 layers.

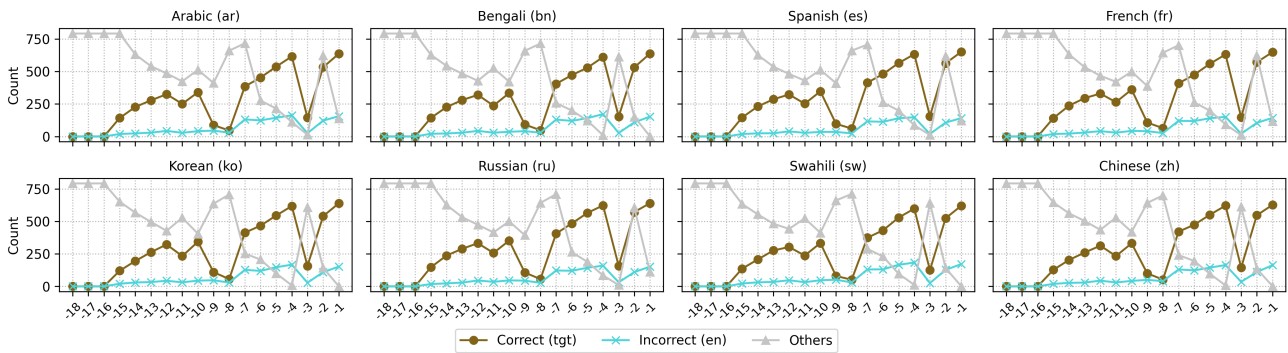

*Figure 16.* **Logit lens visualization per language for GEMMA-3 27B (62 layers).** $x$-axis: Last layer index; $y$-axis: Statement count. We show the last 18 layers to capture the entire pattern.

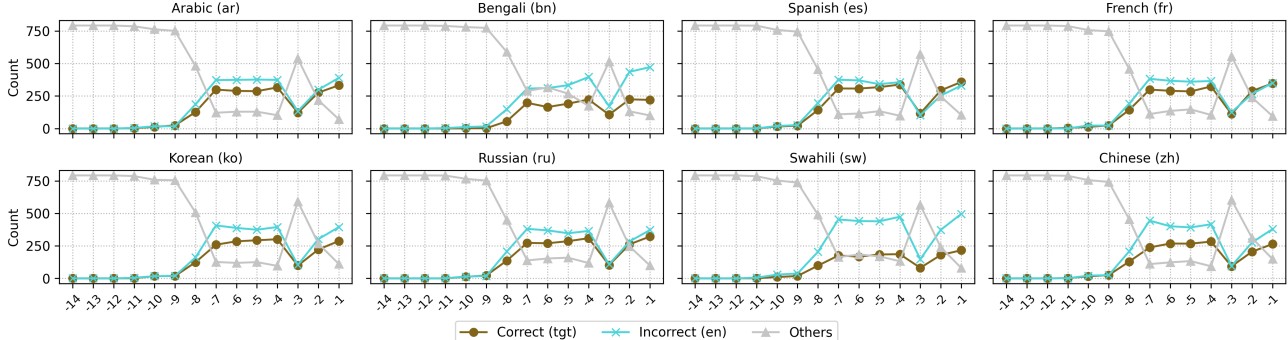

*Figure 17.* **Logit lens visualization per language for AYA23 8B (32 layers).** $x$-axis: Last layer index; $y$-axis: Statement count. We show the last 14 layers.

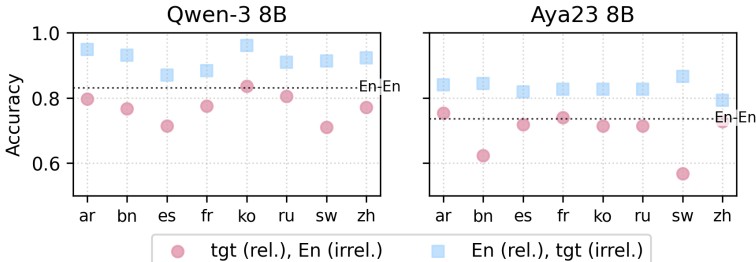

*Figure 18.* **Accuracy per model with one relevant and one irrelevant evidence document in different languages.** ●: Relevant document in target language, irrelevant document in English; ■: Relevant document in English, irrelevant document in target language.

| Model | Language | $d_c = \ell, d_{\neg c} = \ell$ (●) | $d_c = \text{en}, d_{\neg c} = \ell$ (■) | $d_c = \text{en}, d_{\neg c} = \text{en}$ (▲) | $d_c = \ell, d_{\neg c} = \text{en}$ (◆) |
|---|---|---|---|---|---|
| LLAMA-3.1 8B | Arabic | 0.616 | 0.572 | 0.557 | **0.658** |
| | Bengali | 0.673 | 0.619 | 0.683 | **0.729** |
| | Spanish | 0.683 | 0.639 | 0.667 | **0.717** |
| | French | **0.716** | 0.713 | 0.674 | **0.716** |
| | Korean | 0.645 | 0.646 | 0.611 | **0.667** |
| | Russian | 0.736 | 0.666 | 0.686 | **0.745** |
| | Swahili | 0.627 | 0.638 | 0.640 | **0.648** |
| | Chinese | 0.607 | **0.631** | 0.579 | 0.610 |
| LLAMA-3.3 70B | Arabic | 0.828 | 0.843 | **0.858** | 0.837 |
| | Bengali | 0.883 | 0.878 | 0.875 | **0.890** |
| | Spanish | 0.875 | **0.886** | 0.877 | 0.883 |
| | French | 0.875 | 0.893 | 0.880 | **0.906** |
| | Korean | 0.866 | **0.893** | 0.880 | 0.857 |
| | Russian | 0.893 | **0.912** | 0.900 | 0.901 |
| | Swahili | 0.902 | 0.902 | **0.904** | 0.879 |
| | Chinese | 0.792 | **0.815** | 0.772 | 0.811 |
| QWEN-3 8B | Arabic | **0.635** | 0.598 | 0.583 | 0.626 |
| | Bengali | 0.605 | 0.560 | **0.650** | 0.632 |
| | Spanish | 0.648 | 0.600 | 0.603 | **0.665** |
| | French | 0.621 | 0.575 | 0.585 | **0.650** |
| | Korean | 0.677 | **0.705** | 0.672 | 0.655 |
| | Russian | **0.710** | 0.686 | 0.648 | 0.673 |
| | Swahili | 0.477 | 0.459 | 0.463 | **0.479** |
| | Chinese | 0.538 | 0.533 | **0.554** | 0.487 |
| QWEN-3 14B | Arabic | 0.832 | 0.773 | 0.787 | **0.843** |
| | Bengali | **0.910** | 0.892 | 0.901 | 0.909 |
| | Spanish | 0.877 | 0.842 | 0.845 | **0.906** |
| | French | 0.883 | 0.857 | 0.868 | **0.908** |
| | Korean | **0.858** | 0.843 | 0.853 | 0.843 |
| | Russian | 0.889 | 0.867 | 0.875 | **0.898** |
| | Swahili | **0.759** | 0.735 | 0.743 | 0.697 |
| | Chinese | 0.744 | **0.765** | 0.737 | 0.730 |
| GEMMA-3 27B | Arabic | 0.823 | 0.808 | 0.814 | **0.859** |
| | Bengali | 0.868 | 0.867 | 0.873 | **0.897** |
| | Spanish | 0.852 | 0.854 | 0.852 | **0.897** |
| | French | 0.863 | 0.845 | 0.861 | **0.898** |
| | Korean | 0.844 | 0.862 | 0.853 | **0.867** |
| | Russian | 0.878 | 0.851 | 0.867 | **0.902** |
| | Swahili | 0.832 | 0.806 | 0.836 | **0.896** |
| | Chinese | **0.811** | 0.792 | 0.779 | 0.792 |
| AYA23 8B | Arabic | 0.464 | 0.443 | 0.475 | **0.516** |
| | Bengali | 0.454 | 0.401 | 0.354 | **0.537** |
| | Spanish | 0.563 | **0.577** | 0.555 | 0.569 |
| | French | 0.551 | 0.574 | 0.575 | **0.578** |
| | Korean | **0.574** | 0.537 | 0.501 | 0.572 |
| | Russian | 0.531 | 0.540 | 0.532 | **0.590** |
| | Swahili | 0.427 | 0.312 | 0.358 | **0.528** |
| | Chinese | 0.453 | 0.460 | **0.492** | 0.488 |

*Table 11.* **Numerical results when the query is in target language.** We report accuracies for four variants per model and language. We use the same shape notation as in Figure 4. Best scores for each row is **bold**.

| Model | Language | En-En | tgt-En (↓) | En-tgt (↑) |
|---|---|---|---|---|
| LLAMA-3.1 8B | Arabic | 0.944 | 0.931 | 0.961 |
| | Bengali | 0.944 | 0.918 | 0.965 |
| | Spanish | 0.944 | 0.913 | 0.970 |
| | French | 0.944 | 0.939 | 0.970 |
| | Korean | 0.944 | 0.952 | 0.974 |
| | Russian | 0.944 | 0.965 | 0.961 |
| | Swahili | 0.944 | 0.974 | 0.961 |
| | Chinese | 0.944 | 0.944 | 0.970 |
| LLAMA-3.3 70B | Arabic | 0.974 | 0.935 | 0.983 |
| | Bengali | 0.974 | 0.944 | 0.987 |
| | Spanish | 0.974 | 0.926 | 0.987 |
| | French | 0.974 | 0.957 | 0.987 |
| | Korean | 0.974 | 0.944 | 0.978 |
| | Russian | 0.974 | 0.957 | 0.987 |
| | Swahili | 0.974 | 0.961 | 0.978 |
| | Chinese | 0.974 | 0.952 | 0.983 |
| QWEN-3 8B | Arabic | 0.831 | 0.796 | 0.948 |
| | Bengali | 0.831 | 0.766 | 0.931 |
| | Spanish | 0.831 | 0.714 | 0.870 |
| | French | 0.831 | 0.775 | 0.883 |
| | Korean | 0.831 | 0.836 | 0.961 |
| | Russian | 0.831 | 0.805 | 0.909 |
| | Swahili | 0.831 | 0.710 | 0.913 |
| | Chinese | 0.831 | 0.771 | 0.922 |
| QWEN-3 14B | Arabic | 0.961 | 0.926 | 0.970 |
| | Bengali | 0.961 | 0.918 | 0.987 |
| | Spanish | 0.961 | 0.922 | 0.974 |
| | French | 0.961 | 0.944 | 0.970 |
| | Korean | 0.961 | 0.918 | 0.974 |
| | Russian | 0.961 | 0.935 | 0.970 |
| | Swahili | 0.961 | 0.896 | 0.974 |
| | Chinese | 0.961 | 0.918 | 0.961 |
| GEMMA-3 27B | Arabic | 0.944 | 0.887 | 0.970 |
| | Bengali | 0.944 | 0.905 | 0.970 |
| | Spanish | 0.944 | 0.862 | 0.974 |
| | French | 0.944 | 0.905 | 0.961 |
| | Korean | 0.944 | 0.905 | 0.965 |
| | Russian | 0.944 | 0.931 | 0.952 |
| | Swahili | 0.944 | 0.887 | 0.961 |
| | Chinese | 0.944 | 0.883 | 0.965 |
| AYA23 8B | Arabic | 0.736 | 0.753 | 0.840 |
| | Bengali | 0.736 | 0.623 | 0.844 |
| | Spanish | 0.736 | 0.719 | 0.818 |
| | French | 0.736 | 0.740 | 0.827 |
| | Korean | 0.736 | 0.714 | 0.827 |
| | Russian | 0.736 | 0.714 | 0.827 |
| | Swahili | 0.736 | 0.567 | 0.866 |
| | Chinese | 0.736 | 0.727 | 0.792 |

*Table 12.* **Numerical results for setup with one relevant and one irrelevant evidence document, in different languages.** Red denotes when **tgt-En** scores are lower than the **En-En** baseline; Green denotes when **En-tgt** scores are higher than the baseline.

To test this, we use an attribution model ContextCite (Cohen-Wang et al., 2024), which estimates the influence of each document on the model's generation. ContextCite is a fitted linear surrogate model that encodes the importance of each source in the context by taking *ablated* contexts $m \in \{0,1\}^K$ as input, where $m_j = 1$ indicates that sentence $j$ is present and $m_j = 0$ indicates that it is masked. The model predicts the ground-truth logit-scaled probability for a given mask $m$ as:

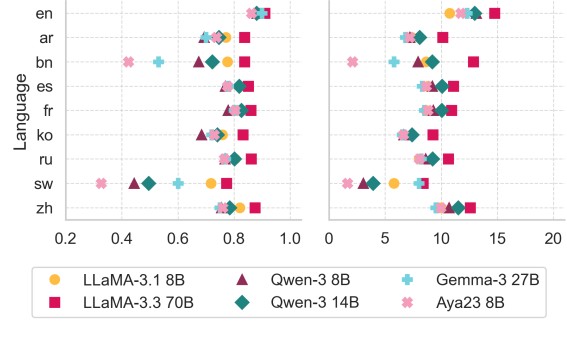

*Figure 19.* **Hit@1 and Score@1 by model and language.** Higher values ($\rightarrow$) indicate more accurate attribution to the cited document.

$$f(m) = w^{\mathsf{T}} m + b, \qquad (4)$$

where $w \in \mathbb{R}^K$ contains per-sentence attribution weights and $b$ is a bias term.

In our case, given a query $q$, a set of $K$ relevant documents $\mathcal{D} = \{d_1, \ldots, d_k\}$, and a pool of statements $\{s_i\}$, ContextCite returns a ranked list of sentences from $\mathcal{D}$ that most influenced the generation of each $s_i$, along with their attribution scores. Here, $\mathcal{D}$ is composed of the cited document in the target language and all remaining documents in English. We evaluate attribution quality using two metrics: (1) **Hit@1** ($\uparrow$): whether the top-ranked sentence originates from the cited document, and (2) **Score@1** ($\uparrow$): the attribution score $w_{j^*}$ of the top-ranked sentence, indicating its estimated relative importance to the model's prediction. Figure 19 presents both metrics by each model and language. Across all models, both metrics peak when the cited document is in English, outperforming all target language counterparts. This suggests that English preference is not merely a surface-level citation pattern but reflects more reliance on English sources during generation.

## F. Language Variants of Non-cited Documents

As described in Section 3.2, our main measurement setup constructs contrastive contexts in which only the document to be cited is in English, while all other documents are in the target language. This design choice isolates the effect of the cited document's language by holding other factors constant—changing all other documents would introduce additional confounders that make cross-language comparison less direct.

To ensure that the observed English preference is not merely an artifact of having the non-cited documents in English, we conduct an additional experiment in which the language of all non-cited documents matches the language of the cited document, while keeping the query in English. Specifically, in **Step 4** (**Next Token Prediction Analysis**) of Section 3.2, we replace the original configuration $\text{Context}(d_{c_i} \rightarrow \ell, d_{\neg c_i} \rightarrow \text{en})$ with $\text{Context}(d_{c_i} \rightarrow \ell, d_{\neg c_i} \rightarrow \ell)$. As shown in Table 13, although citation accuracy increases relative to the original setup (i.e., where non-cited documents are in English), accuracies in this new configuration still remain significantly below the English baseline. This demonstrates that English preference persists even under matched-language contexts.

## G. Constrained Decoding Results

While we carefully control our prompt templates (Appendix A), some valid generations may still fall outside our main accuracy metric—for instance, citation IDs expressed in different languages or stylistic variants. To address this, we conduct an additional experiment using constrained decoding (Post & Vilar, 2018), restricting the model to generate only one of the valid citation ID numbers for each query. This setup removes all stylistic variation. As in Table 14, English still achieves the highest citation accuracy, indicating that the English preference persists even when stylistic differences are fully eliminated.

## H. Qualitative Results

In Table 15, we show qualitative example where the relevant document is provided in its original language and the irrelevant document in English and the model cites the irrelevant one.

| Model | Language | Acc. $(d_\neg c_i = \text{en})$ $(\%, \uparrow)$ | Acc. $(d_\neg c_i = \ell)$ $(\%, \uparrow)$ |
|---|---|---|---|
| LLAMA-3.1 8B | English | 67.4 | 67.4 |
| | Arabic | 59.5** | 61.8* |
| | Bengali | 56.6*** | 59.2** |
| | Spanish | 62.1* | 63.5 |
| | French | 62.9 | 64.3 |
| | Korean | 61.7* | 62.7* |
| | Russian | 62.1* | 63.2 |
| | Swahili | 53.0*** | 57.8*** |
| | Chinese | 59.9* | 58.9** |
| LLAMA-3.3 70B | English | 85.9 | 85.9 |
| | Arabic | 67.3*** | 77.4*** |
| | Bengali | 68.8*** | 78.6* |
| | Spanish | 76.0*** | 80.3* |
| | French | 77.4*** | 80.6* |
| | Korean | 69.2*** | 75.9*** |
| | Russian | 74.5*** | 78.5** |
| | Swahili | 67.3*** | 76.3*** |
| | Chinese | 74.1*** | 75.7*** |
| QWEN-3 8B | English | 62.6 | 62.6 |
| | Arabic | 47.6*** | 51.1*** |
| | Bengali | 41.3*** | 46.8*** |
| | Spanish | 51.9*** | 54.6*** |
| | French | 48.4*** | 50.3*** |
| | Korean | 49.7*** | 55.7** |
| | Russian | 50.4*** | 54.6*** |
| | Swahili | 30.4*** | 39.2*** |
| | Chinese | 49.2*** | 50.4*** |
| QWEN-3 14B | English | 83.0 | 83.0 |
| | Arabic | 72.6*** | 73.8*** |
| | Bengali | 65.4*** | 71.5*** |
| | Spanish | 77.4* | 79.3* |
| | French | 76.0*** | 76.9** |
| | Korean | 70.3*** | 73.8*** |
| | Russian | 74.8*** | 78.3* |
| | Swahili | 54.7*** | 62.8*** |
| | Chinese | 73.5*** | 74.4*** |
| GEMMA-3 27B | English | 86.2 | 86.2 |
| | Arabic | 78.4*** | 79.9** |
| | Bengali | 77.9*** | 80.3** |
| | Spanish | 80.2** | 82.0 |
| | French | 79.0** | 81.4* |
| | Korean | 77.5*** | 80.7* |
| | Russian | 77.1*** | 79.4** |
| | Swahili | 74.0*** | 78.8*** |
| | Chinese | 75.4*** | 79.4** |
| AYA23 8B | English | 60.0 | 60.0 |
| | Arabic | 43.2*** | 49.2*** |
| | Bengali | 27.2*** | 48.3*** |
| | Spanish | 49.1*** | 52.9* |
| | French | 48.5*** | 53.1* |
| | Korean | 42.2*** | 47.4*** |
| | Russian | 48.1*** | 51.7** |
| | Swahili | 22.4*** | 28.9*** |
| | Chinese | 46.3*** | 48.3*** |

*Table 13.* **Citation accuracies (%) when changing the language of non-cited documents.** Pairwise two-sided $t$-tests are performed to compare accuracy between English and the target language, with the null hypothesis that the mean citation accuracy is equal across languages. Bonferroni correction is applied for multiple comparisons. *: significant with $p < 0.05$; **: $p < 0.01$; ***: $p < 0.001$; non-marked: not statistically significant. $d_\neg c_i$: non-cited documents; $\ell$: target language.

| Language | LLAMA-3.1 8B | QWEN-3 8B | AYA23 8B | QWEN-3 14B | GEMMA-3 27B | LLAMA-3.3 70B |
|---|---|---|---|---|---|---|
| **English** | 82.5 | 80.8 | 81.7 | 77.5 | 86.7 | 51.0 |
| **Arabic** | 61.2 (-21.3)*** | 60.5 (-20.3)*** | 65.9 (-15.8)*** | 61.2 (-16.3)*** | 80.2 (-6.5)*** | 41.3 (-9.7)*** |
| **Bengali** | 59.3 (-23.2)*** | 63.5 (-17.3)*** | 63.0 (-18.7)*** | 52.7 (-24.8)*** | 81.4 (-5.3)*** | 36.4 (-14.6)*** |
| **Spanish** | 70.5 (-12.0)*** | 67.4 (-13.4)*** | 72.0 (-9.7)*** | 69.1 (-8.4)*** | 83.5 (-3.2)* | 47.5 (-3.5)* |
| **French** | 71.0 (-11.5)*** | 66.7 (-14.1)*** | 70.3 (-11.4)*** | 68.2 (-9.3)*** | 83.0 (-3.7)** | 47.0 (-4.0)** |
| **Korean** | 65.3 (-17.2)*** | 65.5 (-15.3)*** | 65.0 (-16.7)*** | 62.8 (-14.7)*** | 78.9 (-7.8)*** | 39.5 (-11.5)*** |
| **Russian** | 69.2 (-13.3)*** | 65.9 (-14.9)*** | 68.8 (-12.9)*** | 64.8 (-12.7)*** | 82.1 (-4.6)*** | 43.8 (-7.2)*** |
| **Swahili** | 57.1 (-25.4)*** | 61.9 (-18.9)*** | 47.2 (-34.5)*** | 43.6 (-33.9)*** | 78.4 (-8.3)*** | 35.2 (-15.8)*** |
| **Chinese** | 68.8 (-13.7)*** | 68.4 (-12.4)*** | 68.9 (-12.8)*** | 63.8 (-13.7)*** | 79.0 (-7.7)*** | 39.0 (-12.0)*** |

*Table 14.* **Citation accuracies (%) using constrained decoding.** We present mean accuracy values $\mathbf{Acc}^{(\ell)}$ with $\Delta(\ell_{\text{target}})$ in subscript. Pairwise two-sided $t$-tests are performed to compare accuracy between English and the target language, with the null hypothesis that the mean citation accuracy is equal across languages. Bonferroni correction is applied for multiple comparisons. *: significant with $p < 0.05$; **: $p < 0.01$; ***: $p < 0.001$; non-marked: not statistically significant. Color coding indicates the magnitude of $\Delta(\ell_{\text{target}})$: largest, second largest, others.

| Language | Query | Relevant Doc. | Relevant Doc. (MT) | Irrelevant Doc. |
|---|---|---|---|---|
| **Korean** | How long is the Omo River? | 그 과정은 일반적으로 남쪽으로 이루어 지지만 약 7 ° N 37 ° 30 'E에서 약 36 ° E에서 서쪽으로의 주요 구부러지면서 5 ° 30 'N까지 남쪽으로 돌면 큰 S- 굽힘을 만들어 남쪽으로 투르카나 호수로 다시 코스를 재개합니다. 에티오피아 중앙 통계청이 발표 한 자료에 따르면 Omo-Bottego River의 길이는 760km입니다. | Its course is generally to the south, however with a major bend to the west at about 7° N 37° 30' E to about 36° E where it turns south until 5° 30' N where it makes a large S- bend then resumes its southerly course to Lake Turkana. According to materials published by the Ethiopian Central Statistical Agency, the Omo-Bottego River is 760 kilometers long. | The Omo River forms through the confluence of the Gibe River, by far the largest total tributary of the Omo River, and the Wabe River, the largest left-bank tributary of the Omo at . Given their sizes, lengths and courses one might consider both the Omo and the Gibe rivers to be one and the same river but with different names. |

*Table 15.* **Qualitative example of relevance vs. language preference.** Models cite the irrelevant English document over the relevant document in the original language. **Relevant Doc. (MT)**: English translation version of the relevant document.

# I. Relevance vs. Query Language Preference

We conduct the same set of experiments from Section 7 in a setting where the query is in a language other than English. We use the same dataset, MIRACL (231 queries) with machine translated queries, relevant, and irrelevant documents. While fixing the query in target language, we vary the language of one relevant and one irrelevant document under three conditions: (1) **tgt-tgt** (●): Both relevant and irrelevant documents are in the target (query) language; (2) **tgt-En** (■): Relevant document in the target language, irrelevant document in English; (3) **En-tgt** (▲): Relevant document in English, irrelevant document in the target language.

Our hypotheses are: (*i*) If citation accuracy in **En-tgt** is lower than **tgt-tgt** or **tgt-En**, it suggests that models trade off relevance for query language preference, citing irrelevant target language documents over relevant English ones, and (*ii*) If citation accuracy of **tgt-En** exceeds **tgt-tgt**, it indicates that models more easily ignore irrelevant English distractors, showing stronger query language preference over English preference.

As shown in Figure 20, results support the first hypothesis: citation accuracies are generally the lowest when the relevant document is in English and the distractor is in the query language, showing that models preferring language over relevance persists for non-English queries. Interestingly, we show that this trend is most evident for (1) lower-resource languages such as Bengali (bn) and Swahili (sw) and (2) models that reported to support all tested target languages (QWEN-3 8B and 14B, GEMMA-3 27B; Table 4). Conversely, we show mixed results for the second hypothesis. The citation accuracies of **tgt-En** and **tgt-tgt** are largely similar, suggesting that English distractors are not necessarily easier at misleading models than those in the target language. Overall, our results imply that models show a consistent preference for the query language over relevance, but the distractor's language matters less when the query is not in English.

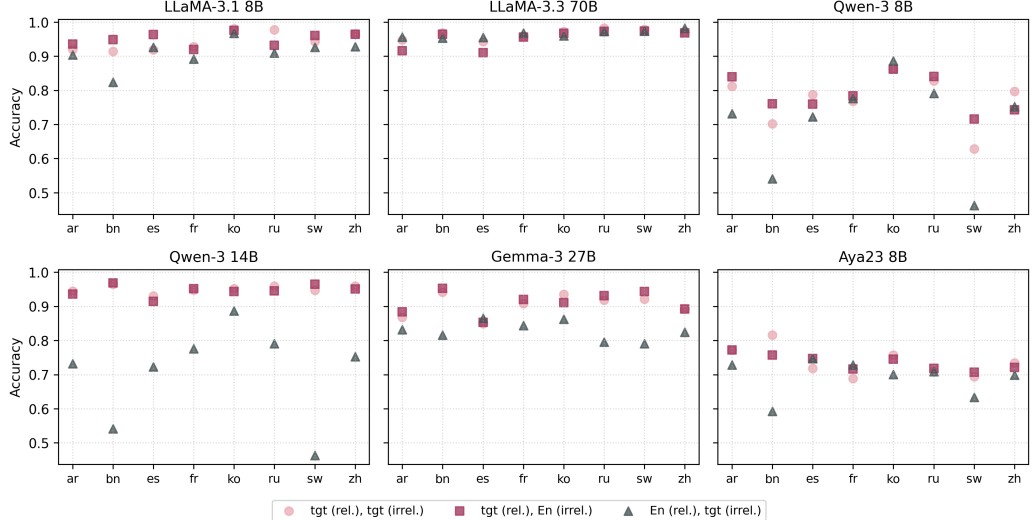

*Figure 20.* **Accuracy per model with one relevant and one irrelevant evidence document in different languages.** ●: Both docs in target language; ■: Relevant doc in target language, irrelevant doc in English; ▲: Relevant doc in English, irrelevant doc in target language. Models also trade off document relevance for language preference for queries not in English.

| Language | LLaMA-3.1 8B | LLaMA-3.3 70B | Qwen-3 8B | Qwen-3 14B | Gemma-3 27B | Aya23 8B |
|---|---|---|---|---|---|---|
| **Arabic** | -0.008 | 0.014 | -0.020 | 0.021 | 0.022 | -0.031 |
| **Bengali** | -0.057 | -0.019 | -0.071 | -0.012 | -0.014 | -0.041 |
| **Spanish** | 0.007 | 0.039 | -0.018 | 0.003 | 0.032 | -0.030 |
| **French** | -0.029 | -0.024 | -0.018 | 0.036 | 0.015 | -0.010 |
| **Korean** | -0.045 | -0.026 | -0.022 | -0.010 | 0.000 | -0.071 |
| **Russian** | 0.001 | -0.057 | -0.040 | -0.002 | -0.007 | -0.034 |
| **Swahili** | -0.028 | 0.014 | -0.002 | 0.025 | 0.029 | -0.083* |
| **Chinese** | -0.030 | -0.040 | 0.014 | -0.039 | 0.005 | -0.026 |

*Table 16.* **Pearson's correlation ($r$) between MT quality of cited document and statement accuracy.** The reported $p$-values correspond to two-sided significance tests for the null hypothesis that the true correlation is zero. *: significant with $p < 0.05$; non-marked: not statistically significant.

## J. Correlation of MT Quality vs. English Preference

We explicitly analyze the relationship between MT quality and English preference. Using COMET-QE scores (Section D.1), we compute segment-level Pearson correlations ($r$) between MT quality and answer accuracy at both (1) the **statement** level: correlating MT quality of the cited document with statement accuracy, and (2) the **query** level: correlating MT quality of the query with its aggregated accuracy. As in Table 16 and Table 17, correlations are consistently none to very weak across all models and languages, indicating no meaningful link between MT quality and English preference.

## K. Additional Experiments with MIRACL

To complement our main experiment results on ELI5, we conduct additional experiments for measuring English preference (§5) and query language preference (§6) on an additional dataset, MIRACL (Zhang et al., 2023).

### K.1. English Preference

We follow the same procedure as in ELI5 (§3) using the English portion of the development set. After **Step 3** (**Statement Pool Construction**), we obtain 818 statements. MIRACL is a non-parallel multilingual RAG dataset—where queries are a mix of long- and short-form and not all evidence documents are required for answering the query. Therefore, results should be interpreted with some caution. Despite these differences, Table 18 shows that English preference observed in

| Language | LLAMA-3.1 8B | LLAMA-3.3 70B | QWEN-3 8B | QWEN-3 14B | GEMMA-3 27B | AYA23 8B |
|---|---|---|---|---|---|---|
| **Arabic** | 0.154* | -0.030 | 0.128* | 0.133* | 0.027 | 0.140* |
| **Bengali** | 0.119 | -0.008 | -0.011 | 0.054 | -0.046 | 0.084 |
| **Spanish** | 0.119 | 0.030 | 0.210 | 0.085 | 0.083 | 0.126 |
| **French** | 0.153* | 0.043 | 0.112 | 0.105 | 0.068 | 0.063 |
| **Korean** | 0.083 | -0.028 | 0.073 | 0.026 | 0.004 | 0.031 |
| **Russian** | 0.097 | -0.030 | 0.142 | 0.033 | 0.034 | 0.035 |
| **Swahili** | 0.183* | 0.144* | 0.120 | 0.155* | 0.076 | 0.052 |
| **Chinese** | 0.195* | -0.001 | 0.058 | 0.028 | -0.068 | 0.083 |

*Table 17.* **Pearson's correlation ($r$) between MT quality of query and aggregated accuracy.** The reported $p$-values correspond to two-sided significance tests for the null hypothesis that the true correlation is zero. *: significant with $p < 0.05$; non-marked: not statistically significant.

| Language | LLAMA-3.1 8B | QWEN-3 8B | AYA23 8B | QWEN-3 14B | GEMMA-3 27B | LLAMA-3.3 70B |
|---|---|---|---|---|---|---|
| **English** | 75.6 | 83.0 | 66.8 | 87.0 | 85.9 | 65.5 |
| **Arabic** | 54.8 (-20.8)*** | 63.5 (-19.5)*** | 41.7 (-25.1)*** | 68.7 (-18.3)*** | 69.3 (-16.6)*** | 48.4 (-17.1)*** |
| **Bengali** | 54.0 (-21.6)*** | 61.4 (-21.6)*** | 38.8 (-28.0)*** | 60.8 (-26.2)*** | 69.3 (-16.6)*** | 32.6 (-32.9)*** |
| **Spanish** | 60.8 (-14.8)*** | 71.6 (-11.4)*** | 49.4 (-17.4)*** | 77.5 (-9.5)*** | 76.0 (-9.9)*** | 53.6 (-11.9)*** |
| **French** | 60.2 (-15.4)*** | 71.0 (-12.0)*** | 47.3 (-19.5)*** | 76.7 (-10.3)*** | 74.6 (-11.3)*** | 54.5 (-11.0)*** |
| **Korean** | 54.7 (-20.9)*** | 62.7 (-20.3)*** | 45.2 (-21.6)*** | 65.9 (-21.1)*** | 67.9 (-18.0)*** | 45.0 (-20.5)*** |
| **Russian** | 58.2 (-17.4)*** | 68.0 (-15.0)*** | 45.7 (-21.1)*** | 70.7 (-16.3)*** | 71.4 (-14.5)*** | 52.7 (-12.8)*** |
| **Swahili** | 52.0 (-23.6)*** | 64.1 (-18.9)*** | 37.0 (-29.8)*** | 59.5 (-27.5)*** | 69.4 (-16.5)*** | 36.2 (-29.3)*** |
| **Chinese** | 56.0 (-19.6)*** | 64.8 (-18.2)*** | 45.4 (-21.4)*** | 70.3 (-16.7)*** | 66.4 (-19.5)*** | 45.6 (-19.9)*** |

*Table 18.* **Citation accuracies (%) using MIRACL.** We present mean accuracy values $\text{Acc}^{(\ell)}$ with $\Delta(\ell_{\text{target}})$ in subscript. Pairwise two-sided $t$-tests are performed to compare accuracy between English and the target language, with the null hypothesis that the mean citation accuracy is equal across languages. Bonferroni correction is applied for multiple comparisons. ***: significant with $p < 0.001$. Color coding indicates the magnitude of $\Delta(\ell_{\text{target}})$: largest, second largest, others.

ELI5 persists in MIRACL, with the English baseline achieving the highest citation accuracy across target languages and models. We further report COMET-QE scores for machine-translated queries, titles, and evidence documents of MIRACL in Table 19, showing reasonable MT quality (average: 0.835 for queries, 0.797 for titles, 0.727 for documents).

### K.2. Query Language Preference

MIRACL also provides non-parallel queries and evidence documents *natively* written in eight target languages. For each language, we randomly sample 100 queries and translate their associated evidence documents into English using Google Translate. We then follow the same procedure as in Section 3. For each supported statement, we compute the next token prediction accuracy while varying only the language of the cited document ($d_c$) to English (en) or the query language ($\ell$). All other variables (query and the non-cited documents remain fixed in the query language). This mirrors the setup in Section 6, with the only change being that MIRACL provides naturally occurring data in target languages. As shown in Table 20, we observe the same query language preference: citation accuracy is consistently higher when $d_c = \ell$ than when $d_c = \text{en}$ across all tested models. This indicates that our findings hold for naturally occurring queries and documents in target languages.

## L. Alternative Machine Translation System

We replicate the main experiments from Section 5 using an alternative machine translation (MT) system to verify whether the English preference trend persists. Specifically, we use TOWER-INSTRUCT 7B[14] (Alves et al., 2024), a model trained for diverse translation-related tasks including general MT, automatic post-editing, and grammatical error correction. Table 22 reports COMET-QE scores by language. Compared to Google Translate (see Table 5), MT quality shows mixed results, with higher scores for all languages except Arabic and Bengali.

---

[14]`Unbabel/TowerInstruct-7B-v0.2`

| Language | COMET-QE$(q, q')$ | COMET-QE$(t, t')$ | COMET-QE$(d, d')$ |
|---|---|---|---|
| **Arabic** | 0.802 | 0.775 | 0.683 |
| **Bengali** | 0.867 | 0.829 | 0.758 |
| **Spanish** | 0.851 | 0.794 | 0.753 |
| **French** | 0.847 | 0.798 | 0.756 |
| **Korean** | 0.862 | 0.814 | 0.741 |
| **Russian** | 0.821 | 0.808 | 0.703 |
| **Swahili** | 0.813 | 0.764 | 0.715 |
| **Chinese** | 0.818 | 0.792 | 0.706 |

*Table 19.* **COMET-QE scores by language for MIRACL.** We evaluate the machine translation (MT) quality of non-English queries $(q)$, titles $(t)$, and evidence documents $(d)$ in the MIRACL dataset. Apostrophe $(')$ indicates MT. Higher scores indicate better MT quality.

Table 23 presents citation accuracies per model and language using TOWER-INSTRUCT translations. We show that the general trend observed with Google Translate persists: models achieve the highest citation accuracy when the cited document is in English. The accuracy gaps between English and target languages are all statistically significant ($p < 0.001$), despite using a stronger MT system compared to Google Translate. This suggests that the English preference cannot be fully attributed to using machine-translated documents. Notably, citing Arabic documents leads to the largest performance drop relative to English across all models, likely reflecting the lower COMET-QE scores for Arabic shown in Table 22.

## M. End-to-end mRAG

We additionally run experiments simulating an end-to-end mRAG framework using NeuCLIR dataset (Lawrie et al., 2025). Specifically, for each English query, we retrieve top-3 documents each for Chinese, Persian, and Russian and compare: (*i*) keeping documents in that language and translating all others into English vs. (*ii*) translating all documents into English. We generate reports with LLAMA-3.3 70B and evaluate them using ARGUE report quality metrics (Walden et al., 2025):

- **Sentence precision:** Proportion of citation-accountable sentences whose citations fully support the sentence, measured on a sentence-level.
- **Nugget recall:** Proportion of reference nuggets (question-answer pairs representing key questions an ideal report should address) that the report correctly answers, measured on a report-level.
- **F1:** Harmonic mean of Sentence precision and Nugget Recall, serving as an overall score for a report, measured on a report-level.

As shown in Table 21 all metrics are consistently higher when all documents are in English, which corroborates the English preference shown by our citation accuracy metric.

| Variant | Sentence precision | Nugget recall | F1 |
|---|---|---|---|
| **Chinese** | 0.279 | 0.049 | 0.080 |
| **Persian** | 0.133 | 0.053 | 0.065 |
| **Russian** | 0.232 | 0.077 | 0.096 |
| **All English** | **0.344** | **0.086** | **0.124** |

*Table 21.* **ARGUE metrics on NeuCLIR dataset.** Chinese, Persian, Russian: keeping documents in respective language and translating all others into English; All English: translating all documents into English. Best scores for each column are **bold**.

## N. Usage of Large Language Models

We used LLMs to support and refine the writing of our work. Importantly, we did not rely on them to generate content or sentences from scratch. Instead, we employed them primarily to polish the clarity and expression of how we presented our results. In addition, we used them for stylistic adjustments, such as improving readability and removing layout issues.

| Language | Model | Acc. ($d_c = $ en) (%, ↑) | Acc. ($d_c = \ell$) (%, ↑) |
|---|---|---|---|
| Arabic | LLAMA-3.1 8B | 45.3 | 65.7 |
| | LLAMA-3.3 70B | 76.6 | 85.3 |
| | QWEN-3 8B | 43.8 | 64.2 |
| | QWEN-3 14B | 70.9 | 83.8 |
| | GEMMA-3 27B | 64.2 | 82.3 |
| | AYA23 8B | 39.3 | 50.6 |
| Bengali | LLAMA-3.1 8B | 50.7 | 72.3 |
| | LLAMA-3.3 70B | 66.2 | 81.1 |
| | QWEN-3 8B | 46.0 | 67.6 |
| | QWEN-3 14B | 71.6 | 89.9 |
| | GEMMA-3 27B | 64.9 | 85.1 |
| | AYA23 8B | 31.1 | 53.4 |
| Spanish | LLAMA-3.1 8B | 60.1 | 65.2 |
| | LLAMA-3.3 70B | 75.0 | 78.4 |
| | QWEN-3 8B | 47.9 | 56.1 |
| | QWEN-3 14B | 75.9 | 80.2 |
| | GEMMA-3 27B | 77.4 | 79.0 |
| | AYA23 8B | 46.7 | 52.7 |
| French | LLAMA-3.1 8B | 68.9 | 75.4 |
| | LLAMA-3.3 70B | 85.8 | 87.7 |
| | QWEN-3 8B | 69.3 | 73.5 |
| | QWEN-3 14B | 80.9 | 90.0 |
| | GEMMA-3 27B | 84.1 | 91.6 |
| | AYA23 8B | 60.2 | 63.1 |
| Korean | LLAMA-3.1 8B | 48.9 | 59.1 |
| | LLAMA-3.3 70B | 69.9 | 76.1 |
| | QWEN-3 8B | 47.7 | 65.9 |
| | QWEN-3 14B | 68.8 | 81.3 |
| | GEMMA-3 27B | 65.3 | 75.6 |
| | AYA23 8B | 44.9 | 64.2 |
| Russian | LLAMA-3.1 8B | 54.3 | 58.4 |
| | LLAMA-3.3 70B | 67.0 | 72.6 |
| | QWEN-3 8B | 49.2 | 65.0 |
| | QWEN-3 14B | 67.5 | 80.2 |
| | GEMMA-3 27B | 67.0 | 77.7 |
| | AYA23 8B | 40.1 | 49.2 |
| Swahili | LLAMA-3.1 8B | 58.0 | 63.0 |
| | LLAMA-3.3 70B | 74.6 | 79.0 |
| | QWEN-3 8B | 44.9 | 52.2 |
| | QWEN-3 14B | 62.3 | 71.0 |
| | GEMMA-3 27B | 58.7 | 74.6 |
| | AYA23 8B | 49.3 | 57.3 |
| Chinese | LLAMA-3.1 8B | 27.8 | 46.3 |
| | LLAMA-3.3 70B | 48.5 | 60.4 |
| | QWEN-3 8B | 35.6 | 38.2 |
| | QWEN-3 14B | 51.9 | 58.2 |
| | GEMMA-3 27B | 45.9 | 63.7 |
| | AYA23 8B | 30.0 | 35.9 |

*Table 20.* **Results when the query is in target language with MIRACL.** $d_c$: cited document; $\ell$: target language. Note that results are comparable only within each target language since MIRACL is a *non-parallel* dataset.

| Language | COMET-QE$(q, q')$ | COMET-QE$(t, t')$ | COMET-QE$(d, d')$ |
|---|---|---|---|
| **Arabic** | 0.467 | 0.374 | 0.311 |
| **Bengali** | 0.802 | 0.562 | 0.491 |
| **Spanish** | 0.855 | 0.595 | 0.574 |
| **French** | 0.859 | 0.598 | 0.548 |
| **Korean** | 0.857 | 0.597 | 0.554 |
| **Russian** | 0.839 | 0.588 | 0.549 |
| **Swahili** | 0.787 | 0.549 | 0.482 |
| **Chinese** | 0.817 | 0.574 | 0.505 |

*Table 22.* **COMET-QE scores by language using TOWER-INSTRUCT.** We evaluate the machine translation (MT) quality of non-English queries ($q$), titles ($t$), and evidence documents ($d$). Apostrophe ($'$) indicates MT. Higher scores indicate better MT quality.

| Language | LLAMA-3.1 8B | LLAMA-3.3 70B | QWEN-3 8B | QWEN-3 14B | GEMMA-3 27B | AYA23 8B |
|---|---|---|---|---|---|---|
| **English** | 67.4 | 85.9 | 62.6 | 83.0 | 86.2 | 60.0 |
| **Arabic** | 24.6 (-42.8) | 21.1 (-64.8) | 15.9 (-46.7) | 26.3 (-56.7) | 26.4 (-59.8) | 23.2 (-36.8) |
| **Bengali** | 45.5 (-22.0) | 52.9 (-33.0) | 34.1 (-28.5) | 53.4 (-29.6) | 52.4 (-33.8) | 38.6 (-21.4) |
| **Spanish** | 58.5 (-8.91) | 72.5 (-13.4) | 50.2 (-12.4) | 73.4 (-9.64) | 74.2 (-12.0) | 47.5 (-12.5) |
| **French** | 53.0 (-14.4) | 65.7 (-20.2) | 41.8 (-20.8) | 66.4 (-16.6) | 65.0 (-21.2) | 42.5 (-17.4) |
| **Korean** | 55.2 (-12.2) | 62.6 (-23.3) | 45.5 (-17.1) | 65.7 (-17.3) | 69.8 (-16.4) | 41.0 (-19.0) |
| **Russian** | 55.1 (-12.3) | 71.6 (-14.3) | 48.9 (-13.7) | 68.1 (-14.9) | 70.1 (-16.1) | 43.3 (-16.7) |
| **Swahili** | 41.7 (-25.7) | 49.5 (-36.4) | 34.2 (-28.4) | 50.9 (-32.1) | 48.9 (-37.3) | 37.1 (-22.9) |
| **Chinese** | 44.2 (-23.2) | 55.8 (-30.1) | 37.4 (-25.2) | 57.7 (-25.3) | 54.8 (-31.4) | 39.3 (-20.7) |

*Table 23.* **Citation accuracies (%) by model and language using TOWER-INSTRUCT 7B translations.** We present mean accuracy values $\text{Acc}^{(\ell)}$ along with $\Delta(\ell_{\text{target}})$ in subscript. Pairwise two-sided $t$-tests are performed to compare accuracy between English and the target language, with the null hypothesis that the mean citation accuracy is equal across languages. Bonferroni correction is applied for multiple comparisons. All differences are statistically significant ($p < 0.001$). Color coding indicates the magnitude of $\Delta(\ell_{\text{target}})$: largest, second largest, others.

