# OpenReview forum: "Linguistic Nepotism: Trading-off Quality for Language Preference in Multilingual RAG"
_ICML.cc/2026/Conference — ICML 2026 spotlight_

### Official Review · Reviewer_u2dm · 2026-03-13

**Soundness:** 3
**Presentation:** 3
**Significance:** 3
**Originality:** 3
**Overall Recommendation:** 4
**Confidence:** 4

**Summary:**

This paper investigates whether RAG systems systematically favor English documents over non-English ones. By translating evidence documents into eight target languages and analyzing the next-token prediction accuracy on citation IDs, the authors demonstrate that six open-weight LLMs disproportionately cite English sources. Crucially, the authors show that models will frequently cite an irrelevant English document over a relevant non-English one.

**Compliance With Llm Reviewing Policy:**

Affirmed.

**Key Questions For Authors:**

- What would happen if you use a different type of RAG, i.e. the typical one where you 'retrieve' from a list of documents by measuring some sort of relevance, Would this findings generalize?
- Some recent models might not need any RAGs for solving ELI5, as you suggest. In contrast, you already explored with MIRACL. Why not use MIRACL (or both to contrast the effect) for all setup?

**Limitations:**

Yes

**Strengths And Weaknesses:**

Strength

I enjoyed reading this paper. The premise is interesting. However, I feel like the flow is not optimal (see weaknesses). Overall, the authors assess a central concept, and the research explores a critical issue for the advancement of equitable language technologies, providing empirical evidence that current models risk reinforcing existing language hierarchies and marginalizing non-English knowledge. Also, the use of logit-lens analysis to reveal that models lock into their citation choices at a specific middle layer provides valuable mechanistic interpretability.

Weakness

Your first two main findings: that English queries prefer English citations, and similarly, target queries prefer target citations are somewhat expected. With how LLMs are designed, they learn to respond in the same language as the query; therefore, I suppose we see similar behavior during RAG. Frankly, I don't see any direct issue with these findings. I would even argue that this is fine behavior, assuming the document quality is identical between languages.

I find that the last finding, where the model prefers an English citation even if it is unrelated, should be the main finding rather than the last. Hence, deeper analysis on this could be explored. This is what the model should not do, and we could see whether it is due to the data. For example, if we ground the main findings on this, then the follow-up ablation can use the same dataset (MIRACL with its translated variants) rather than seemingly two separate experiments, one set with ELI5 and another set with MIRACL.

Perhaps lastly, there are many ways of doing RAG. The motivation is not strong enough on why using the setting used here.

---

> ### Author Rebuttal · Authors · 2026-03-30
>
> We sincerely thank the reviewer for their insightful feedback, for finding the paper enjoyable to read and the problem important, and for carefully articulating concerns about the paper’s flow. We respond point by point.
>
> > **W1+Q2. I would recommend grounding the main findings with Section 7 results and reorganize paper flow. Why not use MIRACL (or both to contrast the effect) for all setup?**
>
> **A1+AQ2.** Thank you for the helpful suggestions! We agree that the behavior highlighted in Section 7 represents the most practically alarming finding (and indeed one of the main motivations behind our title). In the revision, we will restructure the introduction and abstract to better foreground this finding while still preserving the controlled-to-realistic progression of the analysis.
>
> Concretely, we view Section 7 as a *natural extension* of the earlier controlled analyses rather than as a separate or secondary result. Sections 5-6 deliberately operate in a controlled setup to cleanly isolate the effect of language without conflating it with document informativeness or retrieval noise. Section 7 then relaxes this assumption to a more realistic setting where retrievers are imperfect and some retrieved documents are irrelevant, showing that the previously measured language preference can in fact override relevance. Without first establishing language preference under the fully relevant, controlled setup, it would be harder to interpret the Section 7 behavior as a targeted trade-off between relevance and language.
>
> This design choice is also closely tied to our dataset selection. ELI5 is, to our knowledge, the only publicly available dataset that provides (i) knowledge-intensive queries requiring long-form, citation-supported answers, and (ii) evidence documents that are all required to fully answer the query, which fits for Section 5-6. In contrast, as we note in Appendix J, MIRACL contains a mix of short- and long-form queries, and its “relevant” documents are not all required to answer each query, making it a better fit for the Section 7 setting where we explicitly need one relevant and one irrelevant document to study relevance-language trade-offs.
>
> We will clarify this rationale in the main text and narrative link between the controlled ELI5 experiments and the MIRACL‑based analysis in Section 7, so that the latter is clearly presented as building on, rather than standing apart from the previous sections.
>
> > **W2+Q1. There are many ways of doing RAG but the current motivation is not strong enough. What would happen if you use a different type of RAG (i.e., the typical one where you "retrieve" from a list of documents by measuring some sort of relevance? Would these findings generalize?**
>
> **A2+AQ1.** Thank you for raising this concern. We would first like to clarify that **our specific RAG setup is directly motivated by a core limitation of prior multilingual RAG evaluations.** Existing approaches on multilingual RAG typically measure language preference from coarse signals that are confounded by document content, relevance, informativeness, and position in input context (Lines 14-24). To address this, we construct an explicitly controlled setting where we can isolate the effect of *only* the language of the evidence documents while holding these other factors fixed. This, in turn, requires starting from a setting where all evidence documents are required for answering the query, which is difficult to guarantee in typical retrieval pipelines.
>
> At the same time, we agree that "typical" RAG is an important setup and do not intend our controlled setup to replace it. To address this, Appendix J reports experiments with MIRACL, where evidence documents are retrieved from Wikipedia using an ensemble of cross-lingual IR systems (BM25, mDPR, and mColBERT), introducing realistic retrieval noise. Here, we observe that our core English and query language preference findings persist. We further complement this with additional end-to-end mRAG experiments on NeuCLIR (see **A2+AQ2** for Reviewer UfSJ), which again shows higher report quality when all documents are in English, highlighting that our conclusions are not limited to the controlled setup.
>
> More broadly, we see the two setups as complementary: the controlled RAG configuration is what allows us to measure language preference and to quantify how it scales with language resource level, document position, and model type, whereas the MIRACL experiments show that these phenomena are not artifacts of the controlled setup but also emerge under a conventional retrieval-based mRAG pipeline. A fully exhaustive study over all possible RAG variants is beyond the scope of this work, but we will clarify this rationale in the paper and emphasize that our methodology is intended as a building block for different retrieval strategies in future work.
>
>
> We would be happy to address any further questions or concerns you may have.

---

> > ### Author Rebuttal · Reviewer_u2dm · 2026-04-05
> >
> > Thank you for the clarification. I'll keep my high score

---

### Official Review · Reviewer_UfSJ · 2026-03-13

**Soundness:** 3
**Presentation:** 3
**Significance:** 2
**Originality:** 3
**Overall Recommendation:** 5
**Confidence:** 3

**Summary:**

The research explores a critical issue in multilingual RAG: whether models’ citation behavior is systematically biased by document language rather than governed purely by relevance. The paper proposes a controlled evaluation framework that constructs parallel evidence documents via translation, generates citation-supported reference reports, filters statement-citation pairs, and then measures language preference. Empirically, across eight languages and six open-source models, the paper finds a robust English citation preference for English queries, shows that this preference is stronger for lower-resource languages and for documents in middle positions, extends the analysis to query-language preference for non-English queries, and further argues that language can outweigh relevance when relevant and irrelevant documents are mixed.

**Compliance With Llm Reviewing Policy:**

Affirmed.

**Final Justification:**

I raised my soundness score from 2 to 3 and overall recommendation from 4 to 5. The rebuttal was helpful, particularly in clarifying the metric definitions and in adding end-to-end multilingual RAG evidence, which addresses my main concern to some extent. I now find the paper’s main finding more convincing for acceptance.

I still have some reservations about how strongly the current end-to-end results support the broader practical-impact claim. The added evidence is directionally supportive, but still somewhat limited (values in `Table 1` are very small), and the paper continues to rely primarily on a controlled synthetic setup. I therefore support acceptance, while noting that the strength and generality of the end-to-end claim may still be worth discussing further with the other reviewers and the AC.

**Key Questions For Authors:**

1. How well does the proposed citation-ID proxy task (next-token prediction) correlate with end-to-end answer quality?
2. Can the authors provide an end-to-end multilingual RAG experiment to validate the practical impact of the observed bias?
3. Are there concrete examples where language preference causes the model to ignore more relevant evidence and produce a worse answer?

**Limitations:**

yes

**Strengths And Weaknesses:**

### Strengths

1. The paper addresses an under-explored but practically meaningful failure mode in multilingual long-form RAG: citation bias induced by language. This is timely and relevant for multilingual knowledge access systems.
2. The paper does more than show “language preference exists.” It provides comprehensive experiments on positional effects, query-language effects, and relevance-vs-language tradeoffs, making the contribution broader and more insightful.
3. The authors explicitly discuss the dependence on ELI5, the use of machine translation to simulate multilingual evidence, and the controlled nature of the setup, which improves the credibility of the paper.

### Weaknesses

1. The paper lacks theoretical attribution and mechanistic interpretability of the observed phenomenon. The paper largely stops at describing and quantifying language preference, with limited insight into why the phenomenon occurs.
2. The setup environment is clever for control but overly simplified: (i) this paper is not evaluating the full mRAG task but a proxy task; (ii) the multilingual corpus is derived with MT systems and include at most two languages upon one query. Therefore, the connection to end-to-end user-facing mRAG performance is inferential.

---

> ### Author Rebuttal · Authors · 2026-03-30
>
> We sincerely thank the reviewer for recognizing the importance of the problem and the breadth of our analyses. We address each concern below.
>
> > **W1. Lacking theoretical attribution and mechanistic interpretability.**
>
> **A1. We would like to clarify that our primary goal is empirical and behavioral: to isolate and quantify when language preference appears in citation decisions under a controlled setup.**
>
> Within this scope, we provide two complementary mechanistic analyses of *how* this preference manifests: (1) logit lens analysis (Section 5.2) reveals a consistent transition region where valid citation IDs emerge and then remain stable and (2) contributive attribution analysis (Appendix E) shows that English documents exert stronger causal influence on generation. A full circuit-level or formal theoretical account is beyond the scope of this work, but we see it as an important direction for future research.
>
> > **W2+Q2. Simplified setup: (1) lack of end-to-end mRAG, (2) MT-derived data, and (3) at most two languages per query.**
>
> **A2+AQ2.** For (1), we additionally ran an end-to-end mRAG experiment using NeuCLIR (https://neuclir.github.io/). For each English query, we retrieve top-3 documents each for Chinese, Persian, and Russian and compare: (i) keeping documents in that language and translating all others into English vs. (ii) translating all documents into English. We generate reports with LLaMA3 70B and evaluate them using ARGUE report quality metrics (https://arxiv.org/abs/2509.26184). As shown in Table 1, all metrics are consistently higher when all documents are in English, corroborating the English preference shown by our citation accuracy metric.
>
> `Table 1.` ARGUE metrics on NeuCLIR
> | Variant | Citation precision | Nugget Recall | F1 |
> | --- | --- | --- | --- |
> | Chinese | 0.279 | 0.049 | 0.080 |
> | Persian | 0.133 | 0.053 | 0.065 |
> | Russian | 0.232 | 0.077 | 0.096 |
> | All English | 0.344 | 0.086 | 0.124 |
>
> For (2), Appendix I shows no meaningful correlation between translation quality and English preference, and Appendix K shows that English preference persists with a stronger MT system, indicating it cannot be fully explained by translation artifacts. For (3), we deliberately limit each context to at most two languages to isolate the effect of a given language without confounding interactions with additional languages.
>
> > **Q1. Correlation of citation prediction task with end-to-end answer quality?**
>
> **AQ1.** Thank you for the question! **We would first like to note that our citation accuracy metric is not intended as a proxy for end-to-end answer quality as trustworthy mRAG needs both well-attributed citations and high-quality answers.**
> Following RAGAS (https://arxiv.org/abs/2309.15217), we view end-to-end answer quality as comprising faithfulness, context relevance, and answer relevance, where our metric already targets the first two (see **AQ2** for Reviewer KawE). Thus, we prompt GPT‑5 to rate answer relevance (how directly the answer addresses the query) on a 5‑point Likert scale and compute Pearson correlations between per‑query citation accuracy and relevance. In Table 2, correlations are consistently none to weak, indicating that citation accuracy is largely *orthogonal* to answer relevance and that an ideal end-to-end metric should include both.
>
> `Table 2.` Correlation citation accuracy vs. answer relevance
>
> *: $p < 0.05$; **: $p < 0.01$; non-marked: not significant
>
> | Lang. | LLaMA3 8B | LLaMA3 70B | Qwen3 8B | Qwen3 14B | Gemma3 27B | Aya23 8B |
> | --- | --- | --- | --- | --- | --- | --- |
> | ar | 0.137 * | 0.067 | 0.007 | 0.092 | 0.016 | 0.140 * |
> | bn | 0.168 ** | 0.003 | 0.031 | 0.053 | 0.008 | 0.053 |
> | en | 0.045 | 0.016 | 0.018 | 0.041 | 0.004 | 0.094 |
> | es | 0.046 | 0.042 | 0.063 | 0.041 | 0.038 | 0.137 * |
> | fr | 0.106 | 0.055 | 0.036 | 0.050 | 0.055 | 0.113 |
> | ko | 0.109 | 0.077 | 0.002 | 0.031 | 0.048 | 0.100 |
> | ru | 0.156 * | 0.035 | 0.005 | 0.082 | 0.066 | 0.137 * |
> | sw | 0.131 * | 0.104 | 0.051 | 0.151 * | 0.080 | 0.049 |
> | zh | 0.086 | 0.052 | 0.024 | 0.055 | 0.034 | 0.112 |
>
> > **Q3. Concrete examples where language preference causes wrong answers?**
>
> **AQ3.** We show an example where models cite an irrelevant English document over a relevant non-English one, leading to incorrect or incomplete responses. We will include more examples in the revised Appendix.
>
> - **Query**: How long is the Omo River?
> - **Irrelevant doc**: The Omo River forms through the confluence of the Gibe River, by far the largest total tributary of the Omo River, and the Wabe River, the largest left-bank tributary of the Omo. Given their sizes, lengths and courses one might consider both the Omo and the Gibe rivers to be one and the same river but with different names.
> - **Relevant doc**: [...] According to materials published by the Ethiopian Central Statistical Agency, the Omo-Bottego River is 760 kilometers long.
>
>
> We would be happy to address any further questions or concerns you may have.

---

> > ### Author Rebuttal · Reviewer_UfSJ · 2026-04-02
> >
> > Thank you for your detailed response.
> >
> > I think the metric description in the rebuttal is currently a bit imprecise, which makes it confusing for me. In the NeuCLIR overview, citation precision is a citation-level relevance metric, i.e., it measures whether the cited documents are relevant to the topic/request. By contrast, in ARGUE / Auto-ARGUE, the main precision-side metric for report quality is sentence precision/support, which operates at the sentence level and asks whether a sentence is supported by its citations. These are related but not interchangeable quantities, and conflating them makes the rebuttal harder to interpret. Therefore, could you please clarify the definition and calculation of **Citation precision**, **Nugget Recall**, and **F1** in `Table 1`?
> >
> > As for the example provided in AQ3, it seems that both the relevant and irrelevant documents are in English. If the relevant document is actually non-English, could you please specify the exact language and provide the document in its original language rather than a translated version? That would be helpful for understanding. Thanks.

---

> > > ### Author Response · Authors · 2026-04-02
> > >
> > > We sincerely thank the reviewer for the follow-up questions! We address each below.
> > >
> > > > **Q1. Clarify the definition and calculation of citation precision, nugget recall, and F1.**
> > >
> > > **AQ1.** We apologize for the confusion. To be precise, "**Citation precision**" in Table 1 refers to the sentence precision/support metric in Auto-ARGUE. We will correct this terminology to "**Sentence precision**" to align with the naming in the Auto-ARGUE paper. We provide detailed definitions and calculations for each metric below:
> > >
> > > | **Metric** | **Level** | **Definition** | **Formula** | **Calculation Description** |
> > > | --- | --- | --- | --- | --- |
> > > | **Citation precision** (*to be changed to **Sentence precision***) | Sentence | Proportion of citation-accountable sentences whose citations fully support the sentence | `Correctly cited sentences / Sentences with citations or requiring citations` | For example, if a report has 10 citation-accountable sentences and 7 have all citations correctly supporting the sentence, then Sentence precision is 7/10 = 0.7. |
> > > | **Nugget Recall** | Report | Proportion of reference nuggets (QA pairs representing key questions an ideal report should address) that the report correctly answers | `Correct nuggets weighted / Total nuggets weighted` | Each nugget has (1) a set of acceptable answers, (2) an aggregator type (AND/OR) controlling answer matching, and (3) an importance label ("vital" or "okay") determining its weight (2 and 1 respectively). In our setup, a nugget is correct if the report provides *exactly* the full set of required answers (using AND aggregator). The final score weights each correct nugget's contribution by its importance label. For example, if a topic has 10 nuggets and the report exactly provides correct answers to 6 of them, then Nugget Recall is 6/10 = 0.6. If 3 of the 6 correct nuggets are "vital" and 3 are "okay," with all 4 incorrect nuggets being "okay," the weighted Nugget Recall is (3×2 + 3×1) / (3×2 + 7×1) = 9/13 ≈ 0.692. |
> > > | **F1** | Report | Harmonic mean of Sentence precision and Nugget Recall, serving as an overall score for a report | `F1 = 2 × (Sentence precision × Nugget Recall) / (Sentence precision + Nugget Recall)` | For instance, if we take Sentence precision to be 0.7 and Nugget Recall as 0.692, the F1-score will be 2 × (0.7×0.692) / (0.7+0.692) ≈ 0.696. |
> > >
> > > > **Q2. Provide qualitative examples in the original language.**
> > >
> > > **AQ2.** Thank you for the suggestion! In addition to the example provided earlier, we show one more example where the relevant document is provided in its original language and the irrelevant document in English and the model cites the irrelevant one.
> > >
> > > | **Lang.** | **Query** | **Relevant doc (Lang.)** | **Relevant doc (English ver.)** | **Irrelevant doc (English)** |
> > > | --- | --- | --- | --- | --- |
> > > | **Korean** | How long is the Omo River? | 그 과정은 일반적으로 남쪽으로 이루어 지지만 약 7 ° N 37 ° 30 'E에서 약 36 ° E에서 서쪽으로의 주요 구부러지면서 5 ° 30 'N까지 남쪽으로 돌면 큰 S- 굽힘을 만들어 남쪽으로 투르카나 호수로 다시 코스를 재개합니다. 에티오피아 중앙 통계청이 발표 한 자료에 따르면 Omo-Bottego River의 길이는 760km입니다. | *Its course is generally to the south, however with a major bend to the west at about 7° N 37° 30' E to about 36° E where it turns south until 5° 30' N where it makes a large S- bend then resumes its southerly course to Lake Turkana. According to materials published by the Ethiopian Central Statistical Agency, the Omo-Bottego River is 760 kilometers long.* | The Omo River forms through the confluence of the Gibe River, by far the largest total tributary of the Omo River, and the Wabe River, the largest left-bank tributary of the Omo at . Given their sizes, lengths and courses one might consider both the Omo and the Gibe rivers to be one and the same river but with different names. |
> > > | **Bengali** | Where was Jeffrey Dunham born? | ডানহাম জন্মগ্রহণ করেছিলেন ১৮ এপ্রিল, ১৯62২ সালে টেক্সাসের ডালাসে। যখন তিনি তিন মাস বয়সে ছিলেন, তখন তিনি রিয়েল এস্টেট মূল্যায়নকারী হাওয়ার্ড ডানহাম এবং তাঁর গৃহকর্মী স্ত্রী জয়েস দ্বারা গৃহীত হন, যিনি তাকে একমাত্র সন্তান হিসাবে একটি ধনী ডালাস পাড়ায় একটি ধর্মপ্রাণ প্রেসবিটারিয়ান পরিবারে বড় করেছিলেন। | *Dunham was born in Dallas, Texas on April 18, 1962. When he was three months old, he was adopted by real estate appraiser Howard Dunham, and his homemaker wife Joyce, who raised him in a devoutly Presbyterian household in an affluent Dallas neighborhood, as an only child.* | Jeffrey Dunham (born April 18, 1962) is an American ventriloquist and comedian who has also appeared on numerous television shows, including "Late Show with David Letterman", "Comedy Central Presents", "The Tonight Show" and "Sonny With a Chance". He has six specials that run on Comedy Central. Dunham also starred in "The Jeff Dunham Show", a series on the network in 2009. |
> > >
> > > We will include these explanations in the Appendix of the revised manuscript. We hope our responses adequately address your concerns, thank you!

---

### Official Review · Reviewer_KawE · 2026-03-13

**Soundness:** 3
**Presentation:** 4
**Significance:** 2
**Originality:** 2
**Overall Recommendation:** 5
**Confidence:** 4

**Summary:**

This work proposes a controlled methodology to evaluate language bias in mRAG systems by isolating the effect of language from document relevance. Authors utilize a synthetic parallel dataset to test the recall of the multilingual citation across various context configurations. They study several open-weight models and show that there is a systemic bias toward citing English sources for English queries: their results show that models will often trade document relevance off for language preference (especially for lower-resource languages).

**Compliance With Llm Reviewing Policy:**

Affirmed.

**Final Justification:**

The rebuttal addressed the questions I had and the authors have committed to including the identified limitations in their work, accordingly I maintain that the work is useful and that my initial scores continue to reflect support for it.

**Key Questions For Authors:**

1. Can it be assumed that the citation accuracy measured at the token prediction form transfers to the citation accuracy in the inline form? Is there any chance that the model’s secondary + inline citations are robustly multilingual if allowed to generate naturally?

2. Is NLI the most robust way to achieve this extraction verification? Would it be possible to corroborate the performance of the NLI-extraction with some end-to-end eval framework like RAGAS to ensure the judges themselves are not skewing the baseline?

**Limitations:**

Yes

**Strengths And Weaknesses:**

Strengths

1. Comprehensive investigation: Several complementary research questions (in addition to the primary language-preference bias one) are explored. Insights on the fidelity of citations is analyzed under positional shifts and varying target language resource levels gives a  generalizable view of the vulnerability these models exhibit.

2. Controlled measurement setup: Confounding attributes do seem to have been accounted for in the design of the experiments (By looking only at the citation token's probability on frozen generated text, for example).

Weaknesses:

1. Section 5.2 - Decision Boundary Analysis might be an artifact of the probe: The conclusion of the Logit Lens analysis is  methodologically limited; It's not clear if the conclusion is derived because the actual semantic decision-making process occurs at the boundary or simply because the literal citation token appears within the vocab space of the model. This seems like an oversimplification of the reasoning required for document attribution at the decision boundary (Peng, Letian, Chenyang An, and Jingbo Shang. "Correlation and Navigation in the Vocabulary Key Representation Space of Language Models." arXiv preprint arXiv:2410.02284 (2024)).

2. Measurement steps are English biasing/Expected to be suboptimal due to translation artifacts: The methodology relies on machine-translated documents which could introduce artificial artifacts that reduce the naturalness of the document in the target language (especially within the lower resource languages). The consequent effect might be that the synthetic parallel document is already lower in quality to the reference English document. Also, deriving evaluation prompts from an English model processing English text might already inflate the model's alignment with English source documents.

---

> ### Author Rebuttal · Authors · 2026-03-30
>
> We sincerely thank the reviewer for their insightful feedback and for recognizing the strengths of our results and setup. We address the concerns below.
>
> > **W1. Logit lens analysis is methodologically limited.**
>
> **A1.** Thank you for raising this point and for the pointer to the paper! We agree that logit lens-style probes operate in the vocabulary space and therefore cannot precisely localize where semantic decisions are made. Our intention in Section 5.2 is to use logit lens as a descriptive tool to show a consistent pattern across models and languages: there is a transition region where valid citation IDs emerge and then remain stable. We believe that the fact that this transition pattern appears consistently across architecturally diverse models and all eight languages suggests it reflects a genuine behavioral regularity rather than a probe artifact. In the revised version, we will cite the suggested paper and soften our phrasing to describe our findings as evidence for such a transition region in citation predictions, rather than as a precise identification of an internal "decision boundary" layer.
>
> > **W2-1. Results may be biased by translation artifacts.**
>
> **A2-1. We show that (1) translation quality has no meaningful correlation with English preference, and (2) English preference persists even with a stronger translation system.**
>
> In Appendix I, we measure Pearson correlations between translation quality and citation accuracy and consistently find none to very weak correlations. In Appendix K, we run our experiments using a stronger translation system, Tower-Instruct, and still observe statistically significant English preference. Together, these results indicate that the observed English preference cannot be fully explained by translation artifacts.
>
> > **W2-2. The setup already inflates alignment with English source documents.**
>
> **A2-2.** To answer this, we additionally analyze cosine embedding similarities (LaBSE) between the statements and source documents. We find that similarities between English statements and English documents (En-En) are not consistently higher than those between the same English statements and non-English documents (En-Non-En) (Table 1). This again indicates that the observed English preference cannot be solely attributed to English statements being more semantically similar to English documents.
>
> `Table 1.` Cosine embedding similarities
> | Lang. | En-En | En-Non-En |
> | --- | --- | --- |
> | ar | 0.579 | 0.584 |
> | bn | 0.579 | 0.586 |
> | es | 0.579 | 0.586 |
> | fr | 0.579 | 0.589 |
> | ko | 0.579 | 0.577 |
> | ru | 0.579 | 0.589 |
> | sw | 0.579 | 0.588 |
> | zh | 0.579 | 0.586 |
>
> > **Q1. Relation between citation accuracy and in-line citation?**
>
> **AQ1.** We can assume that citation accuracy in our token prediction setup is directly informative for in-line citation accuracy, since we let models freely predict the next token over the full vocabulary and take the top-1, exactly as in standard generation.
> At the same time, our setup is theoretically more robust than evaluating free-form in-line citations since it mitigates the challenge in prior works by restricting to statements whose supporting document has been independently verified, then measure citation accuracy only on these verified pairs, reducing the influence of hallucinated or spurious citations.
>
> > **Q2. NLI corroborate with end-to-end eval framework performance?**
>
> **AQ2. Our new end-to-end evaluation supports that the NLI-based verification closely aligns with a RAGAS-style framework.**
>
> Thank you for this question! In response to the reviewer’s concern, we implement an end-to-end evaluator inspired by RAGAS that uses a single LLM-as-judge (OpenAI o3) to rate each statement for (1) context relevance (which document best supports it) and (2) faithfulness (whether the statement is faithful to the cited document), retaining only statements that pass both. We compare this to our original pipeline (LLM-relevance-judge + NLI) in terms of precision (% of statements retained by the end-to-end evaluator, also retained by our original method), recall (vice versa), and F1. In Table 2, we observe a substantial overlap between the two evaluators across languages, with an overall F1 of 0.847. This supports that our NLI-based verification is robust and well aligned with an end-to-end RAGAS-style evaluator. Additionally, our human evaluation (Appendix C) shows strong agreement between human judgments and our original pipeline.
>
> `Table 2.` Comparison of original vs. end-to-end evaluator
>
> | Lang. | Precision | Recall | F1 |
> | --- | --- | --- | --- |
> | ar | 0.774 | 0.874 | 0.821 |
> | bn | 0.843 | 0.855 | 0.849 |
> | en | 0.836 | 0.845 | 0.840 |
> | es | 0.843 | 0.906 | 0.873 |
> | fr | 0.847 | 0.900 | 0.873 |
> | ko | 0.801 | 0.860 | 0.829 |
> | ru | 0.821 | 0.888 | 0.853 |
> | sw | 0.802 | 0.892 | 0.832 |
> | zh | 0.869 | 0.855 | 0.850 |
> | Overall | 0.826 | 0.875 | 0.847 |
>
> We would be happy to address any further questions or concerns you may have.

---

> > ### Author Rebuttal · Reviewer_KawE · 2026-04-03
> >
> > Thank for answering my questions and running additional experimental that further strengthen your claim. I especially appreciate AQ2 which assuages my concerns about the robustness of the extraction.
> >
> > I wanted to highlight though that A2-2 doesn't necessarily resolve my concern potentially because the chosen model (LaBSE) is by design a language-agnostic model. Since the non-English documents are machine-translated versions of the English source, then similar LaBSE cosine scores mainly show cross-lingual semantic alignment under this script-invariant, language-prior invariant encoder (which I wasn't questioning in the weakness/question I raised).
> >
> > This does not, however, establish that the translated documents are equally natural or free of translation artifacts (backtranslation literature might be useful to probe here). That said, I cannot think of a principal way of ascertaining these artifacts so I will maintain my previous notion of the paper's utility.

---

> > > ### Author Response · Authors · 2026-04-04
> > >
> > > We thank the reviewer for the thoughtful follow-up. We understand and agree that our A2-2 response does not fully address the reviewer’s original concern on whether the translated documents are equally natural or free of translation artifacts. As the reviewer correctly notes, LaBSE's language-agnostic design means high cosine similarity reflects cross-lingual semantic alignment rather than lexical naturalness or the absence of translation artifacts, and we will explicitly acknowledge this distinction in the revised paper. We will also discuss translation artifacts more broadly with our A2-2 response in the Limitations section.
> > >
> > > On more principled ways to understand translation artifacts, we brainstormed several complementary approaches the reviewer might find worth considering:
> > > - **Backtranslation consistency**: Building on the reviewer’s suggestion, backtranslating the machine-translated documents into English and comparing against the original English documents via (a) embedding cosine similarity as a semantic proxy, or (b) Jensen Shannon Divergence over vocabulary distributions [1] as a lexical proxy, could quantify how much meaning and word choice is changed through the translation. We acknowledge, however, that backtranslation introduces its own artifacts [2], so this is best interpreted as a relative signal across languages rather than an absolute quality measure.
> > > - **Perplexity under monolingual language models**: Measuring perplexity of the machine-translated documents using language-specific models (e.g., CroissantLLM [3] for French translations) could detect unnatural phrasing or disfluency. The main limitation is that perplexity scores are not directly comparable across languages, work less effectively in long context settings [4], and suitable monolingual models may not exist for lower-resource languages.
> > >
> > > We recognize that no single approach definitely resolves the concern, and we welcome the reviewer's thoughts on which of these directions they consider most informative or promising. Thank you!
> > >
> > >
> > > **References**
> > >
> > > [1] Guo et al., Do Large Language Models Have an English "Accent"? Evaluating and Improving the Naturalness of Multilingual LLMs (https://aclanthology.org/2025.acl-long.193/)
> > >
> > > [2] Agrawal et al., Quality Estimation via Backtranslation at the WMT 2022 Quality Estimation Task (https://aclanthology.org/2022.wmt-1.54/)
> > >
> > > [3] Faysse et al., CroissantLLM: A Truly Bilingual French-English Language Model (https://arxiv.org/abs/2402.00786)
> > >
> > > [4] Fang et al., What is Wrong with Perplexity for Long-context Language Modeling? (https://arxiv.org/abs/2410.23771)

---

### Decision · Program_Chairs · 2026-04-30

**Decision:**

Accept (spotlight)

**Comment:**

This work proposes a controlled methodology to evaluate language bias in mRAG systems by isolating the effect of language from document relevance. Authors utilize a synthetic parallel dataset to test the recall of the multilingual citation across various context configurations. They study several open-weight models and show that there is a systemic bias toward citing English sources for English queries: their results show that models will often trade document relevance off for language preference (especially for lower-resource languages).


The paper is well motivated and all experiments are sound. There are very few minor concerns left which could potentially be addressed for the camera ready version.